# Chemical Language Models for Natural Products: A State-Space Model Approach

## Abstract

Language models are increasingly applied in scientific domains such as chemistry, where chemical language models (CLMs) are well-established for predicting molecular properties or generating de novo compounds for small molecules. However, Natural Products (NPs)—such as penicillin, morphine, and quinine, which have driven major breakthroughs in medicine—have received limited attention in CLM research. This gap limits the potential of NPs as a source of new therapeutics. To bridge this gap, we develop Natural Product–specific CLMs (NPCLMs) by pre-training the latest state-space model variants, Mamba and Mamba-2, which have shown great potential in modeling information-dense sequences, and compare them with transformer baselines (GPT). Using the largest known collection of $\sim$1M NPs, we provide the first extensive experimental comparison of selective state-space models (S6) and transformers in NP-focused tasks, along with a comparison of eight tokenization strategies, including character-level, Atom-in-SMILES (AIS), general byte-pair encoding (BPE) and NP-specific byte-pair encoding (NPBPE). Model performance is evaluated on two tasks: molecule generation, measured by validity, uniqueness, and novelty, and property prediction (peptide membrane permeability, taste, and anti-cancer activity), evaluated using Matthews Correlation Coefficient (MCC) and Area Under the Receiver Operating Characteristic Curve (AUC-ROC). The results show that Mamba consistently generates 1–2% more valid and unique molecules than Mamba-2 and GPT, while making 3-6% less long-range dependency errors; however, GPT produces $\sim$2% more novel structures. In property prediction, both Mamba and Mamba-2 outperform GPT by a modest but consistent 0.02 to 0.04 improvement in MCC under random splitting. Under stricter scaffold splitting, which groups molecules by core structure to better assess generalization to new scaffolds, all models perform comparably. In addition, chemically informed tokenization further enhances performance. For comparison, we include general-domain CLMs (ChemBERTa-2 and MoLFormer) and found that pre-training on $\sim$1M NPs achieves results on par with general CLMs trained on datasets over 100 times larger, emphasizing the value of domain-specific pre-training and data quality over scale in chemical language modeling.

## 1 Introduction

Natural Products (NPs) are compounds naturally produced by living organisms. Their structural diversity and bioactivity make them particularly valuable candidates for drug discovery (Firn, 2009; Dias et al., 2012; Ertl & Schuffenhauer, 2008). Unlike the common use of 'natural products' to describe plant-based or unmodified consumer goods, here 'NPs' specifically refer to chemical compounds (e.g., caffeine, morphine, and quinine) synthesized by living organisms with ecological or therapeutic relevance. NPs occupy a distinct chemical space shaped by evolution, optimized for biological interaction, and rich in scaffolds rarely found in synthetic compounds (Firn, 2009). Nearly one-fourth of approved drugs between 1981 and 2014 originated from — or were inspired by — NPs, particularly for treating cancer, infectious diseases, and diabetes (Newman & Cragg, 2016). Despite a historical surge in NP discovery, the field has stagnated due to their synthetic complexity, unknown binding targets, and regulatory challenges; however, the potential of NPs in drug discovery

continues to drive interest in new computational methods to explore their largely uncharted chemical space (Harvey et al., 2015; Pye et al., 2017; Saldívar-González et al., 2022).

Advances in artificial intelligence, particularly deep learning and Natural Language Processing (NLP), have enabled new approaches to modeling chemical structures. Representations like SMILES (Simplified Molecular Input Line Entry System) also allow molecules to be treated as text, forming the foundation of CLMs (Weininger, 1988). It is important to note that macromolecules like proteins and DNA have dedicated models and representations. CLMs handle small to medium molecules and their explicit chemical structure in terms of atoms as the basic unit of modelling. CLMs have shown success in molecular generation and property prediction, initially using recurrent architectures like long short-term memory (LSTM) and more recently, transformers (Saldívar-González et al., 2022). Transformers offer strong performance via self-attention but suffer from quadratic complexity, limiting their efficiency on longer or information-dense sequences (Bran & Schwaller, 2024; Bajorath, 2023; Sultan et al., 2024; Jablonka et al., 2024; Vaswani et al., 2017; Tay et al., 2022).

To the best of our knowledge, until now, only causal models, such as GPT, LSTM, and, more recently, structured state-space sequence models (S4), have been applied to NP tasks. We aim to further explore state-space models (SSMs), motivated by prior findings that suggest their superior ability to model complex molecular properties for de novo design compared to GPT (Özçelik et al., 2024). Specifically, we investigate the most recent variants of SSMs, Mamba and Mamba-2, namely selective SSM (S6), that attend to input elements "selectively" over time (Gu & Dao, 2024). Unlike transformers, Mamba process sequences through latent dynamical systems instead of pairwise attention, and they have demonstrated strong performance on a range of sequence modeling tasks, suggesting their potential to effectively capture the complex structural dependencies present in NP molecules (Gu & Dao, 2024; Brazil et al., 2024; Xu et al., 2024; Thoutam & Ellsworth, 2024; Sgarbossa et al., 2024; Peng et al., 2024). NPs are structurally more complex and information-dense than fully synthetic molecules (Firn, 2009; Ertl et al., 2008; Chen et al., 2022; Özçelik et al., 2024; Stahura et al., 2000; Godden & Bajorath, 2001). For instance, Stratton et al. (2015) reported that approved NP drugs average 4.1 nitrogen and 9.3 oxygen atoms per molecule, compared to 2.4 and 2.6 in fully synthetic drugs. NPs also show greater structural complexity, with 8.2 stereocenters and 11 rotatable bonds on average, versus 0.8 and 5.2 in synthetic drugs, underscoring their richer topological and functional diversity. In addition, compared to natural languages, chemical languages lack punctuation and stop words, thus encoding more information per token, which poses more modeling challenges (Ucak et al., 2023).

We use NPs as a case study to probe into selective SSMs versus transformers in modeling dense symbolic sequences, raising hypotheses that may extend beyond chemistry. To this end, we pre-train models from scratch on the largest collection of unique NPs to date, without augmentation by SMILES enumeration, and benchmark Mamba variants against widely used GPT models to address three key questions: (1) Do Mamba models outperform transformer baselines in NP generation and property prediction? (2) Which tokenization strategies are most effective for these tasks? (3) Does fine-tuning CLMs pre-trained on larger and more generic datasets outperform NP-only pretraining? In doing so, we aim to uncover how model type, tokenization, and data-splitting protocols influence model performance on NP sequence processing, while providing reproducible benchmarks and practical insights for future empirical research in this domain.

## 2 RELATED WORK

Most current CLMs are transformer-based and pre-trained on large molecular databases like ZINC (version 22; ∼55B molecules), PubChem (∼119M), and ChEMBL (∼2.5M), with the latter two focused on experimentally characterized compounds (Liao et al., 2024; Bran & Schwaller, 2024; Sultan et al., 2024). These databases mostly contain small molecules, which include natural, semi-synthetic, and synthetic compounds. Despite the importance of NPs in drug discovery, only a few CLMs are explicitly developed for them. Existing NP-focused CLMs often use transformer or recurrent architectures, and more recently S4. They have mostly been trained on datasets like COCONUT, an open-access collection of approximately 700K NPs, and augmented with SMILES enumeration (a technique that generates multiple valid SMILES strings for the same molecule) to generate NP-like molecules (Shen et al., 2024; Sakano et al., 2024; Tay et al., 2023; Özçelik et al., 2024). These

studies show that large language models (LLMs) can effectively generate NP-like molecules. However, no research has yet applied the latest SSMs variants, Mamba and Mamba-2, to NP-focused CLMs or thoroughly investigated tokenization strategies for NPs. This work aims to address these gaps.

# 3 METHODOLOGY

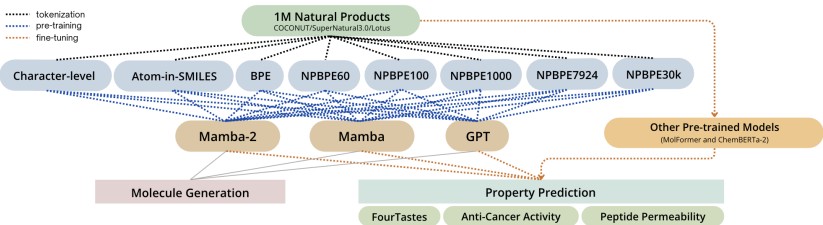

Figure 1: Workflow overview.

## 3.1 DATA

**Pre-training data** used in this study is a curated dataset of 1,030,273 NP SMILES (later referred to as 1M NPs) collected from SuperNatural 3.0, COCONUT, and LOTUS, publicly available databases that compile NPs from commercial sources, open-access repositories, and curated literature (Gallo et al., 2022; Sorokina et al., 2021; Rutz et al., 2022). Duplicate molecules were removed, and the ones flagged by RDKit[1] for the following structural issues were also eliminated: explicit valence issues, kekulization problems, sanitization errors, ambiguous stereochemistry, and hydrogen atoms without neighbors. Subsequently, the remaining SMILES strings in the dataset were standardized and canonicalized using RDKit's `rdMolStandardize`[2] module to remove small fragments, normalize functional groups, and reionize molecules. Charge neutralization is then applied to adjust charged species, followed by tautomer canonicalization to ensure consistent tautomeric forms.

**Downstream data** includes three classification datasets: **(1) Peptide membrane permeability** contains 6,651 cyclic natural peptides with binary membrane permeability labels, curated from CycPeptMPDB using PAMPA assay data (Feller & Wilke, 2025; Li et al., 2023a). **(2) FourTastes** includes 4,431 compounds labeled with one of four classes: "sweet," "bitter," "umami," or "other," curated from ChemTastesDB, UMP442, and other literature (Androutsos et al., 2024). **(3) Anti-cancer activity** consists of ~26,000 compounds with binary labels for general anti-cancer activity, based on growth inhibition data from the NCI-60 screening program (Li & Huang, 2012; Al-Jarf et al., 2021).

All overlapping molecules between 1M NPs and the downstream data have been removed, and all downstream data underwent the same preprocessing as the 1M NPs. Table 5 and Figure 9 in the Appendix show downstream task dataset size, class, and feature distributions. These three datasets were chosen because peptides represent a subset of NPs, while the other two capture common biological roles of NPs, such as shaping taste and smell perception and serving as a rich source of anti-cancer agents (Firn, 2009). Therefore, FourTastes and anti-cancer activity contain a relatively high proportion of NP molecules, although they are not composed exclusively of NPs. Purely NP-specific datasets are either not publicly available or have not been established, primarily due to the high costs associated with conducting assays on a sufficiently large number of NPs necessary for machine learning applications (see Appendix A.3.2 for details).

## 3.2 MODEL

The Mamba model used in this study is the `MambaLMHeadModel` [3], a wrapper around the Mamba architecture with a language modeling head. Both Mamba and Mamba-2 share the same overall

---

[1] https://www.rdkit.org/

[2] https://www.rdkit.org/docs/source/rdkit.Chem.MolStandardize.rdMolStandardize.html

[3] https://github.com/state-spaces/mamba/blob/main/mamba_ssm/models/mixer_seq_simple.py#L215

architecture and differ only in the mixer component. The main difference between Mamba and Mamba-2's neural network architecture is that the SSM parameters in Mamba-2 are produced in parallel with the input instead of sequentially as in Mamba (Dao & Gu, 2024). The generative pre-trained transformer (GPT) model used in this study is `GPT2LMHeadModel` [4]. It is a decoder-only model with a language modeling head from the GPT-2 family. All models are pre-trained from scratch on A100 GPUs, with architectures and configurations detailed in Appendices A.4 and A.12.

## 3.3 TOKENIZERS

This study employs eight tokenizers spanning character-level, atom-level, and motif-level methods. These include a simple character-level tokenizer (vocab size: 48), an atom-level Atom-in-SMILES (AIS) tokenizer (vocab size: 1023) designed to encode atomic environments (Ucak et al., 2023), and six byte-pair encoding (BPE) variants. One BPE tokenizer is adopted directly from DeepChem's Hugging Face model `seyonec/PubChem10M_SMILES_BPE_450k` [5] with a vocabulary of 7,924 tokens, though only around 1,700 are used for 1M NPs. The remaining five BPE tokenizers (NPBPE) are trained on 1M NPs using Hugging Face's `BpeTrainer` [6], with vocabulary sizes of 60, 100, 1,000, 7,924, and 30k. All tokenizers (except DeepChem's) were implemented using a part of the prebuilt functionalities from Hugging Face's `PreTrainedTokenizer` [7] with fixed vocabulary files built from the 1M NPs. Tokenizer implementation details are provided in Appendix A.6, with examples of tokenized SMILES strings, Jaccard similarity between tokenizer vocabularies, and token Zipfian distributions shown in Appendix Table 7 and Figures 14 and 15.

## 3.4 EXPERIMENTAL DESIGN

The study begins by pre-training 48 model variants (from Mamba, Mamba-2, and GPT with eight tokenizers using both random and scaffold splits) on the 1M NPs split into training (80%), validation (10%), and test (10%) sets. Scaffold split separates molecules with the same scaffolds or core chemical structures into the same set, helping to better evaluate models' ability to generalize to structurally novel compounds. The scaffold splitter used in this study is DeepChem's `ScaffoldSplitter` [8] class, which is based on the Bemis-Murcko scaffold representation.

**Pre-training** hyperparameters are optimized via random search on 5% of the training/validation data. SMILES sequences are truncated or padded to a fixed length of 512 tokens and trained with cross-entropy loss using the optimized hyperparameters for each model/tokenizer pair until convergence ($\leq$150 epochs with early stopping, patience = 5 epochs). Afterwards, they are evaluated on an unseen test set. The total training time for each model ranges from hours to a week. The parameter counts for each model range from 8.96M to 69.7M (see Appendix A.12 for details). All models are trained with Sharpness-Aware Minimization (SAM), adopting the implementation from Tsai et al. (2024). SAM is an optimization technique that encourages convergence to flatter loss minima, which are associated with better generalization performance (Foret et al., 2020).

**Fine-tuning** is conducted on three downstream tasks using the 48 pre-trained models and two additional models, MoLFormer and ChemBERTa-2 (Ross et al., 2022; Ahmad et al., 2022), which undergo a two-step fine-tuning process: with or without being fine-tuned on the 1M NPs first before being fine-tuned for downstream tasks. MoLFormer is an encoder-only transformer-based CLM that learns chemical representations from SMILES using rotary positional embeddings, linear attention, and Regex-based tokenization; in this study, we use the `MoLFormer-XL-both-10pct` variant, which has been pre-trained on a 10% subset ($\sim$100M) of PubChem and ZINC, and performs comparably to the full-scale `MoLFormer-XL` (Ross et al., 2022). ChemBERTa-2 is a RoBERTa-based transformer pre-trained on 77M SMILES from PubChem; in this study, we fine-tune the MLM vari-

---

[4]`https://huggingface.co/docs/transformers/v4.51.3/en/model_doc/gpt2#transformers.GPT2LMHeadModel`

[5]`https://huggingface.co/seyonec/PubChem10M_SMILES_BPE_450k`

[6]`https://huggingface.co/docs/tokenizers/en/api/trainers#tokenizers.trainers.BpeTrainer`

[7]`https://huggingface.co/docs/transformers/v4.51.3/en/main_classes/tokenizer#transformers.PreTrainedTokenizer`

[8]`https://deepchem.readthedocs.io/en/latest/api_reference/splitters.html#scaffoldsplitter`

ant on 1M NPs, while using the more computationally intensive MTR variant directly for property prediction (Ahmad et al., 2022). Details can be found in Appendices A.4.3 and A.4.4.

**Property prediction** is performed by replacing each model's language modeling heads with classification heads. Each task is evaluated via repeated 5×5-fold cross-validation with hyperparameter tuning, early stopping, and results are averaged over runs with standard errors (SE) (Figures 16 and 17 in the Appendix). Loss function for the FourTastes dataset is `CrossEntropyLoss()` with class weighting since it is an imbalanced multi-class task, while `BCEWithLogitsLoss()` with-

| | | Downstream Tasks | | | | | | |
|---|---|---|---|---|---|---|---|---|
| | | Property Prediction | | | | | | Unconditional |
| **Models** | Pre-training*/ Fine-tuning** on 1M NP Data Split | Peptide Permeability (AUC-ROC ± SE) | | FourTastes (MCC ± SE) | | Anti-Cancer Activities (MCC ± SE) | | Molecule Generation (novel, valid, unique NP-Likess, SAScore) |
| | | scaffold | random | scaffold | random | scaffold | random | |
| **NP Pre-trained Models** (current study, 48 models) | scaffold* | ✓ | ✓ | ✓ | ✓ | ✓ | ✓ | ✓ |
| | random* | ✓ | ✓ | ✓ | ✓ | ✓ | ✓ | ✓ |
| **MolFormer** (MoLFormer-XL-both-10pct) | x | ✓ | ✓ | x | x | ✓ | ✓ | |
| | random** | ✓ | ✓ | x | x | ✓ | ✓ | |
| **ChemBERTa-2** (ChemBERTa-77M-MLM) | x | ✓ | ✓ | x | x | ✓ | ✓ | n.a. |
| | random** | ✓ | ✓ | x | x | ✓ | ✓ | |
| **ChemBERTa-2** (ChemBERTa-77M-MTR) | x | ✓ | ✓ | x | x | ✓ | ✓ | |

Figure 2: Downstream application overview.

out weighting is applied to the more balanced anti-cancer and peptide permeability binary prediction tasks. Evaluation metrics include Area Under the Receiver Operating Characteristic Curve (AUC-ROC) and Matthews Correlation Coefficient (MCC). **Molecule generation** is performed via autoregressive sampling with a temperature of one and a maximum sequence length of 512 tokens. Each of the 48 pre-trained models generates 100k SMILES sequences, and they are assessed for validity (% of chemically valid SMILES strings, as determined by RDKit), uniqueness (% of non-duplicate valid molecules), and novelty (% of valid, unique molecules not found in 1M NPs). Figures 1 and 2 provide an overview of the experimental design and downstream task application. All pre-trained models, implementation code, and generated pseudo-NP SMILES in this work will be made publicly available via Hugging Face and GitHub upon acceptance.

# 4 RESULTS

## 4.1 MOLECULE GENERATION

Model and tokenizer choices give rise to three general trends. First, tokenizers with smaller or chemically informed vocabularies (e.g., Character-level, AIS, NPBPE60/100) outperform larger vocabulary ones in validity, uniqueness, and novelty by around 10-20%. Second, Mamba yields 1-2% higher validity and uniqueness, while GPT generates around 2% more novel molecules and scaffolds. Third, tokenizer choice significantly impacts performance.

Table 1: **Tokenizer- and model-wise average percentages of valid, unique, and novel molecules:** values for random and scaffold split pre-trained models and their average; "novel" refers to not being present in the 1M NP training data.

| | Random Split | | | Scaffold Split | | | Average | | |
|---|---|---|---|---|---|---|---|---|---|
| | Valid | Unique | Novel | Valid | Unique | Novel | Valid | Unique | Novel |
| *Tokenizer-wise Avg.* | | | | | | | | | |
| Character-level | 80.68% | 79.85% | 70.23% | 79.26% | 78.17% | 66.94% | 79.97% | 79.01% | 68.58% |
| AIS | **81.43%** | **80.25%** | **70.41%** | **81.09%** | **79.73%** | **68.85%** | **81.26%** | **79.99%** | **69.63%** |
| BPE | 78.09% | 77.12% | 66.66% | 78.79% | 77.55% | 65.31% | 78.44% | 77.34% | 65.99% |
| NPBPE60 | 80.33% | 79.34% | 68.10% | 79.05% | 77.97% | 68.03% | 79.69% | 78.65% | 68.06% |
| NPBPE100 | 80.60% | 79.46% | 67.85% | 81.02% | 79.45% | 67.06% | 80.81% | 79.45% | 67.46% |
| NPBPE1000 | 80.81% | 79.14% | 64.30% | 79.65% | 77.36% | 61.06% | 80.23% | 78.25% | 62.68% |
| NPBPE7924 | 68.50% | 66.39% | 53.18% | 69.04% | 66.25% | 51.14% | 68.77% | 66.32% | 52.16% |
| NPBPE30k | 57.29% | 54.94% | 45.24% | 50.25% | 47.43% | 37.92% | 53.77% | 51.19% | 41.58% |
| *Model-wise Avg.* | | | | | | | | | |
| Mamba-2 | 74.70% | 73.42% | 61.80% | 74.51% | 72.69% | 58.90% | 74.60% | 73.05% | 60.35% |
| Mamba | **77.25%** | **75.91%** | 63.11% | **75.28%** | **73.64%** | 61.35% | **76.27%** | **74.77%** | 62.23% |
| GPT | 75.95% | 74.36% | **64.83%** | 74.52% | 72.64% | **62.11%** | 75.23% | 73.50% | **63.47%** |

**Tokenizer impact** on generation performance shows that AIS achieves the highest molecule validity, uniqueness, and novelty, on average 1-2% higher than the character-level and small vocabulary NPBPE variants, which perform second best (see Table 1). In contrast, larger NPBPE tokenizers (e.g., NPBPE7924/30k) show substantially degraded performance, around 12-28% lower on average than AIS, potentially due to excessive vocabulary size. This aligns with the findings from Aksamit et al. (2024), which shows that overly large vocabularies lead to sparse token usage and hinder efficient chemical language modeling.

Table 2 shows that tokenizers with more fine-grained tokens, like character-level and NPBPE60, however, yielded more unique and novel scaffolds (Murcko scaffolds extracted from the generated molecules using RDKit's `GetScaffoldForMol`[9]), contrasting with the earlier observation that the AIS tokenizer is more effective for molecules. Scaffolds may benefit from finer-grained tokenization because they involve core substructures that recur with subtle variations, and fine-grained tokens offer such flexibility. Scaffold sets generated by the 48 models also show low pairwise Jaccard similarity values (from 0.05 to 0.11), suggesting that different models are exploring fairly distinct chemical spaces (Figure 23 in the Appendix). The scaffold distribution in the 1M NPs follows Zipf's law, and so do the scaffolds from the generated molecules (Figure 22 in the Appendix). Novel scaffolds predominantly appear in the long tail of the distribution, typically occurring fewer than 10 times each, reflecting their low frequency and high diversity.

**Model Influence** on molecule generation is more subtle compared to the influence of tokenizers. However, on average, Mamba still yields 1.4% higher validity and uniqueness than Mamba-2 and GPT. In contrast, GPT outperforms Mamba and Mamba-2 by approximately 2% on average in generating novel molecules and has generated about 1-2k more novel scaffolds (see Tables 1 and 2). Mamba's selective SSM (S6) architecture is less effective than transformers in forming entirely novel molecules, likely due to its

Table 2: **Tokenizer- and model-wise average unique and novel scaffold counts:** values for random and scaffold split pre-trained models and their average.

| | Random Split | | Scaffold Split | | Average | |
|---|---|---|---|---|---|---|
| | Unique | Novel | Unique | Novel | Unique | Novel |
| *Tokenizer-wise Avg.* | | | | | | |
| Character-level | **43641** | **33757** | 34000 | 25134 | 38821 | 29446 |
| AIS | 40014 | 29184 | 32306 | 22377 | 36160 | 25781 |
| BPE | 41372 | 31972 | 33750 | 24742 | 37561 | 28357 |
| NPBPE60 | 42637 | 32001 | **36323** | **28406** | **39480** | **30204** |
| NPBPE100 | 42825 | 32338 | 34793 | 25892 | 38809 | 29115 |
| NPBPE1000 | 41385 | 29461 | 31183 | 21060 | 36284 | 25261 |
| NPBPE7924 | 35500 | 25957 | 28172 | 19867 | 31836 | 22912 |
| NPBPE30k | 30183 | 23133 | 20971 | 15811 | 25577 | 19472 |
| *Model-wise Avg.* | | | | | | |
| Mamba-2 | 39313 | 29067 | 30315 | 21266 | 34814 | 25166 |
| Mamba | **40355** | 29394 | 31878 | 23016 | **36117** | 26205 |
| GPT | 39415 | **30716** | **32119** | **24452** | 35767 | **27584** |

local, sequential update and the hidden state decay in its global channels, as discovered by Ye et al. (2025), which together limit the holistic understanding required for novelty. In contrast, self-attention's attending to all tokens at each step likely enables more flexible token recombination than Mamba. Mamba outperforms Mamba-2 in producing more valid, unique, and novel molecules. This aligns with the expected trade-off in Mamba-2 between inference expressivity and hardware efficiency; namely, by replacing a diagonal with a scalar-times-identity structure in its SSM, Mamba-2 reduces expressivity, while Mamba retains more flexible, independently controlled channels (Gu & Dao, 2024; Dao & Gu, 2024; Dao, 2024).

**Training and inference efficiency** varies by model architecture and tokenizer. Larger-vocabulary NPBPE tokenizers reduce the average tokenized sequence length by approximately 3- to 6-fold (from around 70 tokens with a character-level tokenizer to 10–20 tokens) and often result in an average 2- to 4-fold increase in training speed, though at the cost of reduced generation quality. Additionally, NPBPE60 and 100 achieve performance nearly as well as Character-level and AIS tokenizers while halving sequence length through only a few simple merges (e.g., "C(", "CC", "O)", "[C", "[C@", "cc"), showing that optimized tokenization can reduces memory and compute while maintaining performance. Furthermore, hyperparameter search also consistently leads to Mamba and Mamba-2 models adopting deeper architectures (with more stacked Mamba blocks) than GPT models, leading to slower training and inference than GPT, which tends to select fewer transformer blocks (Figures 28, 29, and 30 in the Appendix). Specific model-tokenizer pairs, like M1+NPBPE100 and GPT+NPBPE100, balance outstanding overall performance and speed, with over 50% faster training and up to 12 times faster inference than the worst-performing ones. GPT+AIS and GPT+NPBPE60 are the most efficient and competitive for novel scaffolds. These results underscore the importance of aligning model-tokenizer choices with specific goals for optimal outcomes.

It is worth noting that while evaluating the synthesizability of machine-generated pseudo-NPs is important, it remains highly challenging. Although advances in AI-assisted synthesis planning (e.g., AiZynthFinder, Chemformer, RXN, Synthia, ASKCOS, ICSYNTH) show promise, these tools re-

---

[9] `https://www.rdkit.org/docs/source/rdkit.Chem.Scaffolds.MurckoScaffold.html`

main limited primarily to simpler or moderately complex molecules, with automated routes for highly intricate, sp³-rich NPs still unreliable and often requiring extensive expert intervention and validation (Gangwal & Lavecchia, 2025). For additional discussion and evaluation on synthetic accessibility, see Appendix A.11. Ultimately, the primary goal of this study is not practical synthesis, but to understand and evaluate how well models and tokenizers learn NP properties and explore their chemical space.

**Error analysis** is therefore conducted by using `partialsmiles 2.0`, a Python library that parses partial SMILES to detect syntax, valence, and kekulization errors (O'Boyle, 2020). Across all models, there is a clear pattern of syntax errors dominating around 80% of the molecules rejected by `partialsmiles`, and nearly half of them arising from unclosed rings (Table 3 and Appendix Table 9). While valence errors reflect local inconsistencies, and kekulization failures stem more from global graph-level constraints, certain syntax errors can provide insight into how well a model handles long-range dependency. As shown in Table 3, Mamba generates 6.15% and 3.73% fewer long-range syntax errors than GPT and Mamba-2, while also producing more valid and unique molecules (Table 1). Mamba is less effective at novelty compared to GPT; nevertheless, Mamba has produced much fewer long-range dependency related errors, and its selective SSM updates provide more structural stability in contrast with GPT's flexible but more error-prone novel token recombinations. Similar robustness has been noted previously, with S4 (the predecessor of Mamba) shown to capture long-distance dependencies and reduce SMILES design errors more reliably than GPT (Özçelik et al., 2024), and Mamba demonstrated to possess Lyapunov-stable recurrent dynamics that prevent divergence under perturbations (Halloran et al., 2025).

Table 3: **Model- and tokenizer-wise average of syntax errors related to long-range dependency.** All values are in % and are calculated by dividing the count of each error type by the total number of molecules rejected by the `partialsmiles` library.

| Syntax Error Type | GPT | M1 | M2 | Char | BPE | AIS | 60 | 100 | 1000 | 7924 | 30k |
|---|---|---|---|---|---|---|---|---|---|---|---|
| N ring openings have not been closed | 48.32 | 47.58 | 44.76 | 42.74 | 46.72 | 48.38 | 44.85 | 45.72 | 48.70 | 51.42 | 46.60 |
| Unmatched close parenthesis | 14.82 | 13.20 | 18.18 | 17.11 | 11.75 | 15.57 | 13.90 | 12.59 | 11.26 | 16.87 | 24.18 |
| N branches have not been closed | 8.62 | 4.30 | 5.84 | 5.33 | 7.62 | 7.41 | 7.78 | 5.56 | 4.13 | 5.57 | 6.63 |
| Missing the close bracket | 0.66 | 0.92 | 0.97 | 1.81 | 2.23 | 0.00 | 1.38 | 0.86 | 0.43 | 0.06 | 0.03 |
| The final branch should not be within parentheses | 0.39 | 0.65 | 0.62 | 0.39 | 0.19 | 0.14 | 0.38 | 0.57 | 0.69 | 0.87 | 1.19 |
| An open square bracket without close bracket | 0.01 | 0.02 | 0.03 | 0.06 | 0.01 | 0.00 | 0.03 | 0.02 | 0.02 | 0.00 | 0.00 |
| Total | 72.82 | **66.67** | 70.40 | 67.44 | 68.52 | 71.50 | 68.32 | 65.32 | **65.23** | 74.79 | 78.63 |

M1 = Mamba, M2 = Mamba-2; numeric labels (60, 100, 1000, 7924, 30k) refer to NPBPE variants with corresponding vocabulary sizes.

## 4.2 PROPERTY PREDICTION

This section introduces the evaluation of 48 NP pre-trained models alongside MoLFormer and ChemBERTa-2 across three downstream property prediction tasks: molecular taste, anti-cancer activity, and peptide membrane permeability prediction. The analysis examines how tokenizer choice, model architecture, fine-tuning, and data splitting influence performance across these tasks.

**Tokenizer choice** has a substantial impact on model performance in downstream tasks, as shown by prior studies across natural, chemical, and protein language modeling (Dotan et al., 2024; Leon et al., 2024; Lindsey et al., 2024; Schmidt et al., 2024; Dagan et al., 2024; Suyunu et al., 2024; Tan et al., 2024). Same as for molecule generation, chemically informed tokenizers such as AIS and small, domain-adapted NPBPE variants (e.g., NPBPE60/100/1000) generally outperform larger vocabulary tokenizers like NPBPE7924 and NPBPE30k, as shown in Figures 3a and 3b. Peptide permeability prediction, however, shows less sensitivity to tokenizer and model variation, particularly under scaffold-split conditions (Figure 3c). This is likely because cyclic peptides have more uniform and constrained structures (Kurita & Numata, 2024). As a result, this task exhibits less variance, making it easier and less sensitive to model and tokenizer choices.

**Impact of data splitting and model selection** is evident from the clear performance drop observed when downstream task data are scaffold split (columns on the right in Figures 3 and 4), which challenges models to generalize by removing structural overlaps between training and test sets. In contrast, performance differences between scaffold and random splits on 1M NPs during pre-training are minimal. Model-wise, conditions of random splitting downstream data tend to highlight

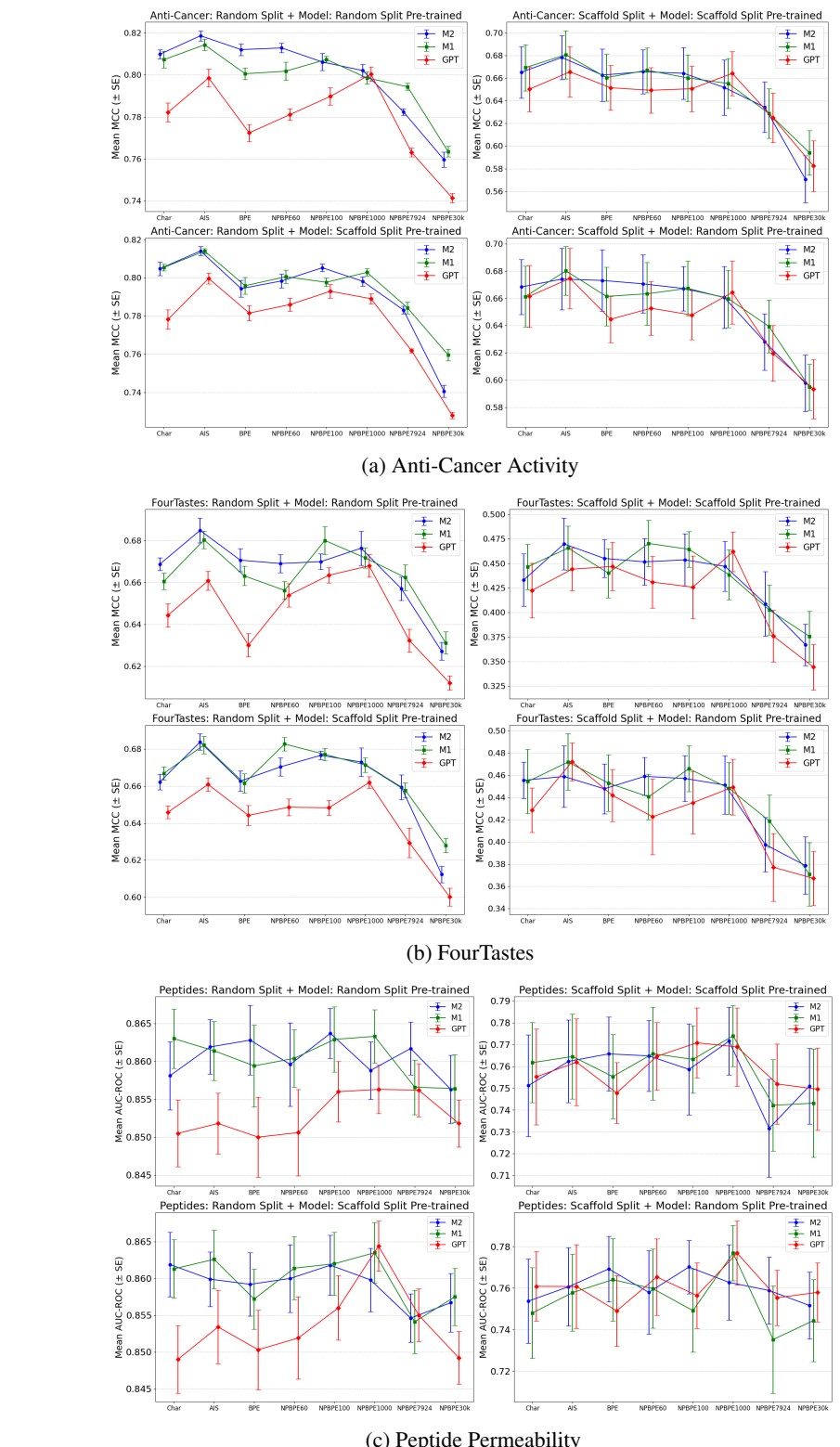

(a) Anti-Cancer Activity

(b) FourTastes

(c) Peptide Permeability

Figure 3: **Model performance comparison**: mean MCC and AUC-ROC (± SE) values across tokenizer and model pairs for three property prediction tasks with random and scaffold data split combinations.

architecture-related differences (columns on the left in Figures 3 and 4). Under random split, both Mamba models generally outperform GPT. A reason might be that Mamba's S6 dynamically prioritizes relevant information within a single sequence processing step to effectively capture local sequential dependencies (Gu & Dao, 2024). Mamba models can use such learned patterns readily and directly under random split conditions, where training and test sets share similar molecular scaffolds. For tasks that require more local focus, such a way of learning local patterns may be more effective than self-attention, which attends to all tokens at every step. Recurrent models have also been reported to provide better learning of local patterns than self-attention, which is better at capturing more global properties (Chen et al., 2023). Since Mamba is built upon SSMs, which are inherently recurrent, it might have also inherited this behavioral feature. Scaffold splitting downstream task data, however, requires models to capture more abstract, global chemical properties that extend beyond local motifs—a scenario in which neither Mamba's local focus nor GPT's inherently global self-attention seems to provide a clear advantage.

**Pre-training vs. fine-tuning** on NPs shows that NP-specific models trained on just 1M NP molecules can match the performance of CLMs (ChemBERTa-2 and MoLFormer) with large-scale pre-training data (77–110 times larger). In the anticancer activity task (Figure 4a), the Mamba model with AIS tokenizer performs nearly on par with MoLFormer under random split. Although MoLFormer (without NP fine-tuning) performs best, the gains are marginal and may not justify the much larger pre-training dataset. For peptide per-

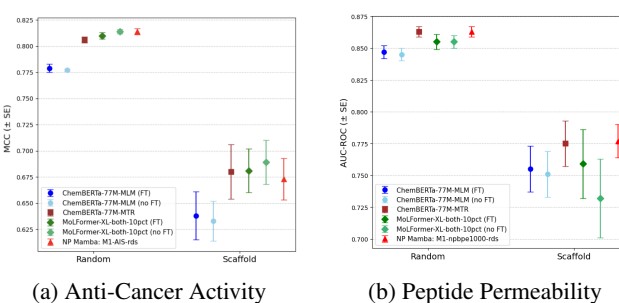

(a) Anti-Cancer Activity  (b) Peptide Permeability

Figure 4: **Model performance comparison**: mean MCC and AUC-ROC (± SE) values for all models with and without fine-tuning (FT) on 1M NPs.

meability (Figure 4b), the NP Mamba model performed almost the same as ChemBERTa-77M-MTR under both random and scaffold split. Additionally, fine-tuning MoLFormer and ChemBERTa-77M-MLM on 1M NPs yields minor improvements. These observations have demonstrated the value of domain-specific pre-training, and they align with findings from other studies, which demonstrate that broad pre-training may introduce noisy or irrelevant features for the usually more specialized tasks in fields such as cheminformatics, bioinformatics, and material science (Kirschbaum & Bande, 2024; Skinnider et al., 2021; Fournier et al., 2024; Ghunaim et al., 2025; Li et al., 2023b). While training domain-specific CLMs from scratch offers greater control over data quality and avoids reproducibility issues, it demands substantial computational resources. Alternatively, Sultan et al. (2025) report that partial pre-training on relevant data (reduced-scale pre-training plus domain adaptation) can yield better performance, highlighting a trade-off between flexibility and efficiency.

## 5    CONCLUSION

This work addresses the underrepresentation of NPs in chemical language modeling by developing NPCLMs using both selective SSMs (Mamba, Mamba-2) and transformer (GPT) architectures. Through a systematic comparison across eight tokenization strategies and two downstream tasks (molecule generation and property prediction), we demonstrate that Mamba generates more valid and unique molecules while making fewer long-range dependency errors. However, GPT excels in novelty. Mamba and Mamba-2 also slightly outperform GPT in property prediction under random splits, though performance aligns under scaffold splits. The tokenization strategy strongly influences the results: chemically informed and domain-specific tokenizers with appropriate token frequency distribution outperform others. Domain-specific pre-training proves to be effective and can match the performance of fine-tuned general CLMs, emphasizing the importance of data relevance and quality over volume. In essence, for information-dense sequences, selective SSMs tend to preserve structural validity and long-range consistency, whereas self-attention favors recombination and novelty. Future work should examine whether these insights extend beyond the domain of NPs, while also aiming to advance tools to assess the synthesizability of machine-generated NP molecules, explore alternative inference and training methods, and investigate model size, architecture, and training data volume trade-offs.

## REPRODUCIBILITY STATEMENT

We have taken comprehensive steps to ensure the reproducibility of our work. **(1) Datasets:** The pre-training and downstream property prediction datasets, together with all preprocessing steps, are described in Section 3.1 of the main text and detailed in Appendix A.3. We have included the random split version of the 1M NP dataset and the first-fold random split of the three downstream tasks in the supplementary material's `data` folder. **(2) Tokenizers:** Implementations of all tokenizers and their vocabulary files are provided in the `tokenisers.py` file and the `vocab_files` directory in the supplementary repository. **(3) Experimental setup:** The design of training, hyperparameter search, and fine-tuning procedures for all models is documented in Section 3.4 of the main text, with full details in Appendices A.7 and A.12. **(4) Code and scripts:** The supplementary repository includes a `Dockerfile` to build the training environment and a `main.py` entry point for running all tasks. A `run_experiments.sh` script is provided as a template to execute pre-training, fine-tuning, hyperparameter search, and molecule generation with minimal configuration; users can uncomment the relevant blocks to reproduce specific experiments. One pretrained model (`M1-npbpe60-rds`) is provided for demonstration. Upon acceptance, all 48 pretrained models, dataset splits, and generated molecules will be released on Hugging Face and GitHub. **(5) Execution instructions:** The supplementary repository's `README` file contains step-by-step instructions to rerun all major experiments reported in the paper, ensuring that results can be reproduced end-to-end without ambiguity.

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

# A  APPENDIX

## CONTENTS

### A.1 LLM Usage Disclosure

We have used ChatGPT (versions 4o and 5) for two minor purposes: (1) to aid with language polishing, and (2) to assist in the retrieval and discovery of related work. The models were not used for idea generation, analysis, or writing substantive parts of the paper.

### A.2 Limitations

#### Synthesis feasibility

Although most generated pseudo-NPs are structurally valid according to RDkit, many can be synthetically infeasible (Gao & Coley, 2020). Existing synthesis tools (e.g., AiZynthFinder, Chemformer, IBM RXN, Merck Synthia, ASKCOS) are designed for small synthetic molecules and perform poorly on the structurally rich, $sp^3$-heavy space of NPs. We therefore acknowledge the limitation of not being able to assess the real-world synthesizability of the generated pseudo-NPs directly. Future collaborations in chemoenzymatic synthesis and protein design (e.g., NRPS for peptides) are needed to bridge this gap between computational generation and experimental synthesis. For completeness, we still report the synthetic accessibility evaluation of generated pseudo-NPs in Section A.11.

#### Data availability

Another limitation is the lack of sufficiently large NP-only property prediction datasets, which can result in constraints in model assessment. While several NP databases exist, many rely on machine learning (ML) model-predicted rather than experimentally validated properties, and those with assay data are often too heterogeneous or fragmented to be useful for ML tasks (see Section A.3.2 for details). This lack of standardized, high-quality datasets limits both model benchmarking and evaluation.

#### Model size and architecture

Finally, the study does not explore how differences in model size, such as those among Mamba, Mamba-2, and GPT, might impact performance; prior work suggests observed differences may partly reflect parameter count disparities (Kaplan et al., 2020; Ross et al., 2022). While our hyperparameter search favored larger Mamba configurations, a broader comparison across CLMs (Table 4) shows that smaller models such as ChemBERTa-77M-MTR achieved competitive results despite having fewer parameters, suggesting that scale alone does not explain performance. More systematic work is needed to disentangle the trade-off between architecture, pre-training dataset size, and parameter count.

Table 4: **Model parameters and pre-training dataset size.** Model parameters are counted by summing all elements in the tensors returned by `model.parameters()`.

| Model | Parameters | Pre-training Dataset Size |
|---|---|---|
| ChemBERTa-77M-MLM | 3.4M | 77M |
| ChemBERTa-77M-MTR | 3.4M | 77M |
| MoLFormer-XL-both-10pct | 44.4M | 110M |
| M1-AIS-rds | 59.1M | 1M |
| M1-npbpe1000-rds | 39.8M | 1M |

### A.3 DATA

#### A.3.1 PRE-TRAINING DATA

The pre-training dataset adopted in this study has been collected from three NP databases: SuperNatural 3.0 (Gallo et al., 2022), COCONUT (Sorokina et al., 2021), and Lotus (Rutz et al., 2022). Duplicate molecules were removed, and the ones flagged by RDKit for the following structural issues were also eliminated: explicit valence issues, kekulization problems, sanitization errors, ambiguous stereochemistry, and hydrogen atoms without neighbors. Subsequently, the remaining SMILES strings in the dataset were standardized and canonicalized using RDKit's `rdMolStandardize`[10] module to remove small fragments, normalize functional groups, and reionize molecules. Charge neutralization is then applied to adjust charged species, followed by tautomer canonicalization to ensure consistent tautomeric forms. These cleaning steps ensure the remaining dataset contains valid and interpretable molecular structures. After removing 4811 molecules that overlap with the downstream task data, the final curated pre-training NP dataset contains 1,030,273 SMILES strings, including around 21% common to all three sources and a notable overlap of almost 54% of the data points shared between COCONUT and Supernatural 3.0. This combined dataset is referred to as the 1M NP dataset throughout this study. More information regarding its feature distribution can be found in Figure 9.

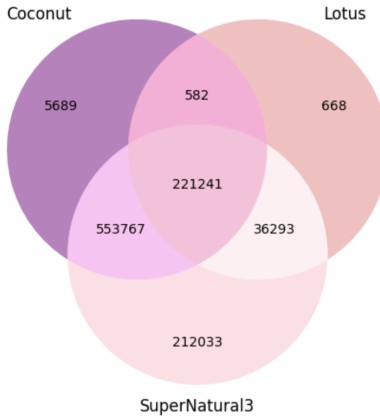

Figure 5: Pre-training data composition Venn diagram.

#### A.3.2 NP DOWNSTREAM DATASET AVAILABILITY AND TASK SELECTION

To the best of our knowledge, few classification datasets exist for NP molecule property prediction. The Cyclic Peptide Membrane Permeability Database (CycPeptMPDB)[11] is one of the few, given that peptides are a subset of NPs. Several NP databases, such as SuperNatural 3.0[12], NPASS[13], and the Natural Products Atlas[14], include properties of NP molecules, but they are primarily predicted values from ML algorithms rather than carefully curated assay results. This is problematic and not ideal in our use case, as using model-predicted feature values as input for new models can propagate existing bias. Other NP databases, such as the Comprehensive Marine Natural Products Database (CMNPD)[15] and Taiwan Database of Extracts and Compounds (TDEC)[16], contain assay results. However, their assay results come from various laboratories for different activity types, cell lines, and targets, and filtering them by the source, activity, cell line, and target renders the dataset

---

[10] https://www.rdkit.org/docs/source/rdkit.Chem.MolStandardize.rdMolStandardize.html
[11] http://cycpeptmpdb.com/
[12] https://bioinf-applied.charite.de/supernatural_3/
[13] https://bidd.group/NPASS/about.php
[14] https://www.npatlas.org/
[15] https://www.cmnpd.org/
[16] https://tdec.cmu.edu.tw/index_en.aspx#

too small for machine learning tasks. Thus, it is not feasible to curate meaningful NP property prediction datasets from these databases.

Nevertheless, based on the common biological activities of NPs, such as being a rich source of anti-cancer drugs and their evolutionary influence on taste and smell perception, which organisms, including humans, have relied on to select food and navigate the environment (Firn, 2009), we have chosen FourTastes and anti-cancer activity, besides the peptide membrane permeability data, as the downstream property prediction tasks. Although most of these datasets do not consist exclusively of NPs, they likely contain a higher proportion of them.

Table 5: Dataset size and label distribution.

| Dataset | Total Data Points | Label Distribution | |
|---|---|---|---|
| Peptide Permeability | 6651 | $\text{LogP}_{\text{exp}} \geq -5.5$ | 2836 (42.64%) |
| | | $\text{LogP}_{\text{exp}} < -5.5$ | 3815 (57.36%) |
| FourTastes | 4431 | Sweet | 1831 (41.32%) |
| | | Bitter | 1776 (40.08%) |
| | | Other | 635 (14.33%) |
| | | Umami | 189 (4.27%) |
| Anti-Cancer | 26039 | Inactive | 14548 (55.87%) |
| | | Active | 11491 (44.13%) |

### A.3.3 PEPTIDE PERMEABILITY

Peptides constitute a subset of NPs. In the 1M NP dataset, peptides account for approximately 2% of the total compounds, with the majority originating from the Lotus database. The peptide permeability classification dataset used in this study was curated by Feller & Wilke (2025) from the Cyclic Peptide Membrane Permeability Database (CycPeptMPDB), a database compiling membrane permeability data for cyclic peptides (Li et al., 2023a). CycPeptMPDB aggregates information on 7,334 cyclic peptides from various papers and pharmaceutical patents. To refine the dataset, Feller and Wilke included only peptides with results from the Parallel Artificial Membrane Permeability Assay (PAMPA), excluding data from cell-based permeability assays. Data points that were categorized as undetectable were also removed to ensure permeability predictions were based on a more reliable and consistent subset of data. The dataset was further processed into a binary classification format using a $10^{-5.5} \log P_{\text{exp}}$ threshold. Peptides with values below this threshold were classified as nonpermeable, while those at or above this value were classified as permeable. The current study has further removed 50 peptides because they overlap with the 1M NP training data, resulting in a final dataset comprising 6,651 cyclic peptides. Despite its small size, it is made of NPs entirely and has a relatively balanced label distribution, as shown in Table 5.

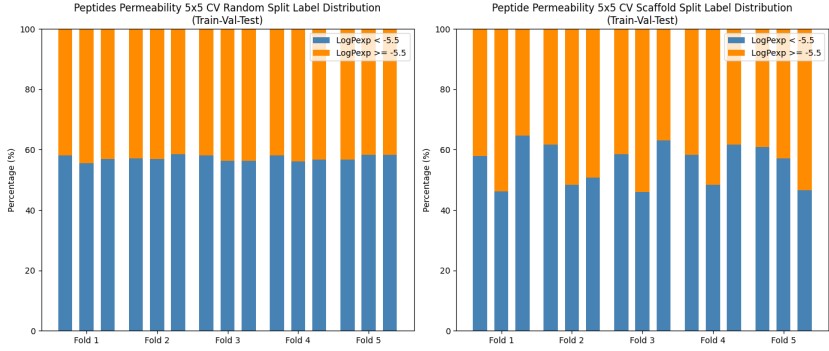

Figure 6: Peptide permeability dataset 5x5 CV random and scaffold split label distribution.

### A.3.4 FOURTASTES

The FourTastes dataset is a tastes prediction dataset curated from UMP442, ChemTastesDB, and various previous literature by Androutsos et al. (2024). The dataset includes four taste labels: "sweet",

"bitter", "umami", and "other". After preprocessing and removing overlapping data points, the final dataset consists of 4,431 data points. This dataset can be comparatively more relevant for NPs, as taste perception can be closely linked to their chemical properties. Natural compounds, such as alkaloids and flavonoids, also contribute to specific taste profiles. For example, caffeine, a kind of alkaloid, is responsible for the bitterness in coffee, while flavonoids like catechins in green tea contribute to astringency and bitter notes.

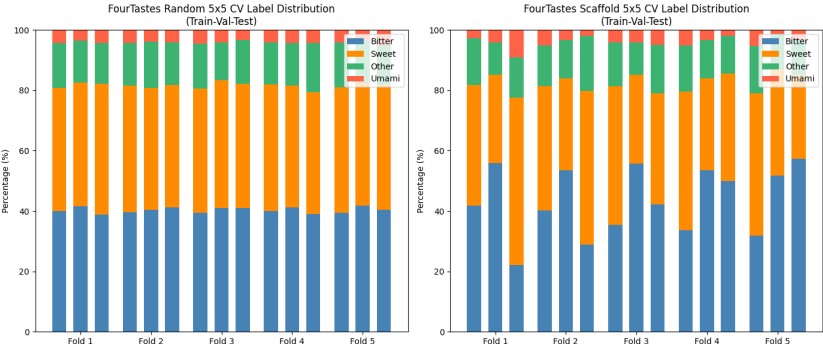

Figure 7: FourTastes dataset 5x5 CV random and scaffold split label distribution.

### A.3.5 ANTI-CANCER ACTIVITY

The anti-cancer activity dataset is a binary classification dataset that differentiates between active and inactive compounds based on their ability to inhibit cancer growth. This dataset contains approximately 28,000 data points. It was initially curated by Li & Huang (2012) and extended by Al-Jarf et al. (2021), using the same curation method. The data used in this study is obtained from the pdCSM-cancer website[17]. The original source of the data is the NCI-60 Development Therapeutics Project (DTP), which screens small molecules against over 60 human cancer cell lines derived from nine tumor types. The bioactivity classification was determined using growth inhibition percentages (GIPRCNT) at a dose of $10^{-5}$M. Compounds were classified as inactive if the average growth inhibition rate was lower than 5% (GIPRCNT>95%) in One Dose assays and active if the average inhibition rate was higher than 50% (GIPRCNT<50%) in Dose Response assays.

This dataset is valuable for its large size and standardized assay conducted by a single institution, ensuring relatively high-quality and reliable experimental results. However, it does not differentiate activity by cancer type. In real-world applications, cancer treatments are usually developed for specific tumor types. While this approach allows for the identification of broad-spectrum anti-cancer compounds, it risks missing those that have strong efficacy limited to particular cancer types.

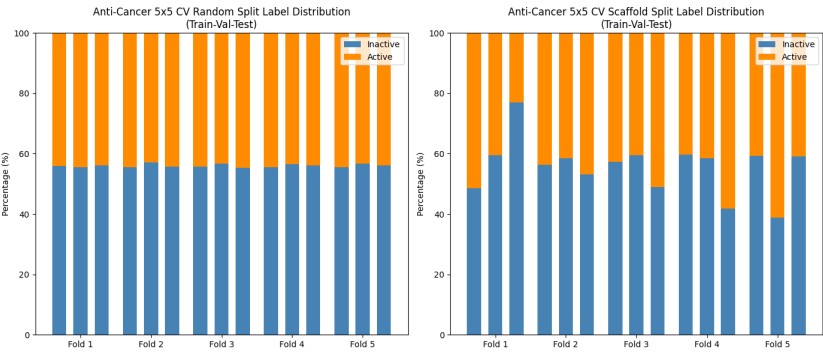

Figure 8: Anti-cancer activity dataset 5x5 CV random and scaffold split label distribution.

---

[17]https://biosig.lab.uq.edu.au/pdcsm_cancer/

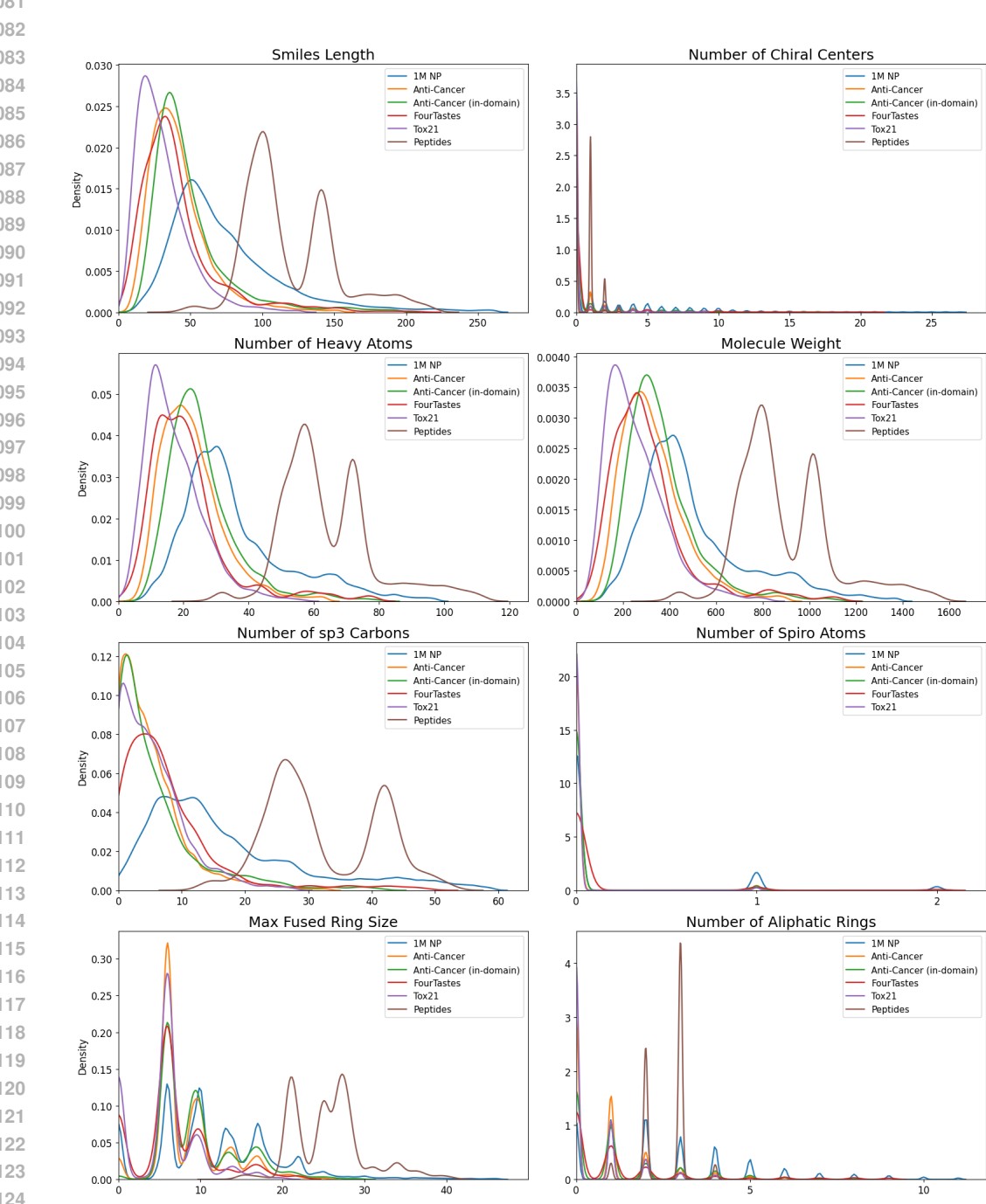

Figure 9: **Kernel density estimation (KDE) plots of molecular features across all datasets.** Features are calculated using RDkit. Extreme values beyond the 99th percentile are excluded, and bandwidth is set to 1 for all features except the number of chiral centers and number of aliphatic rings, for which a smaller bandwidth of 0.5 is used. Only assigned chiral centers are included. No spiro atoms are present in the peptide permeability dataset. Integer-valued features may show KDE curves at non-integer positions due to kernel smoothing, which spreads density beyond discrete points but does not imply fractional values.

## A.4 MODELS

### A.4.1 GPT

The generative pre-trained transformer (GPT) model used in this study is `GPT2LMHeadModel`. It is a decoder-only model with a language modeling head from the GPT-2 family. It is specifically for causal sequence generation. In this study, the GPT-2 model is trained from scratch without any pre-trained weights. GPT-2's attention layer uses masked multi-head self-attention to process sequences. Self-attention allows each token to attend to previous ones, and multi-head attention splits attention into multiple subspaces for richer feature extraction. Masked (causal) attention ensures that tokens can only attend to past and present tokens, preventing information leakage from future tokens. This structure enables GPT-2 to generate text autoregressively.

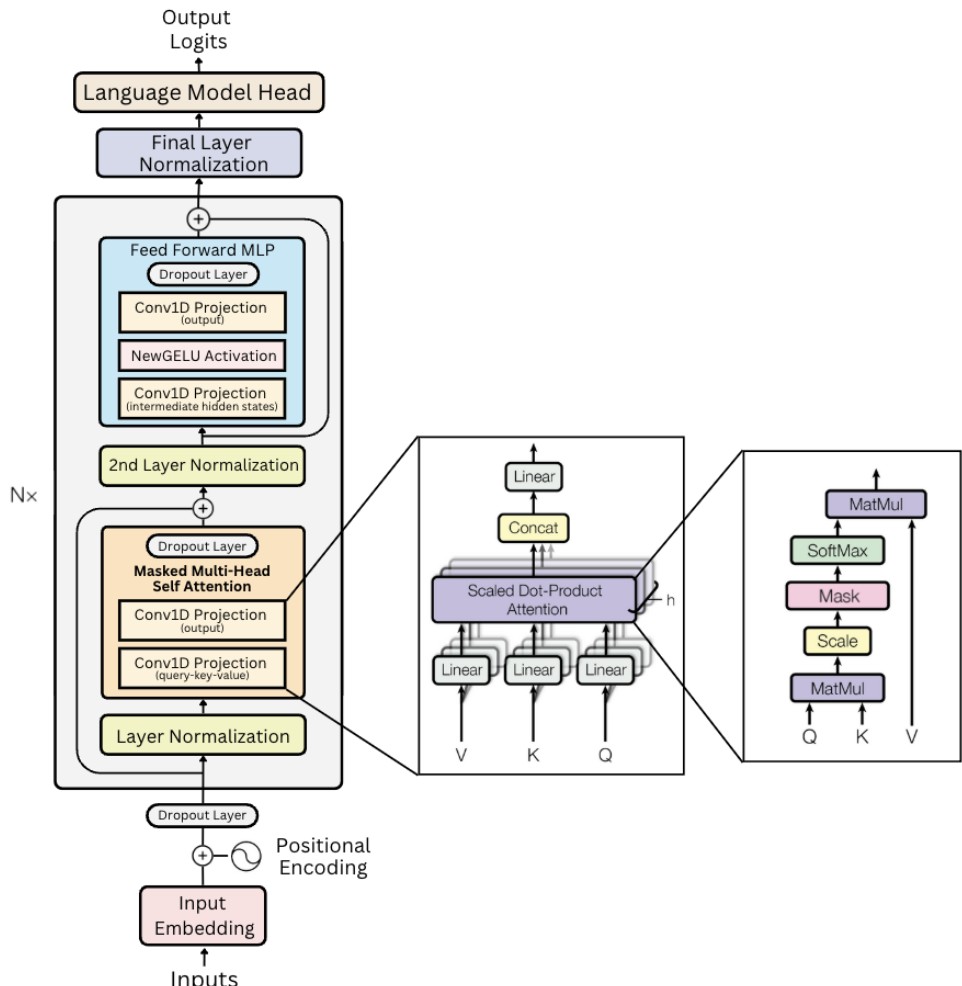

Figure 10: GPT language model architecture (adapted from Vaswani et al. (2017)).

### A.4.2 MAMBA AND MAMBA-2

The Mamba model used in this study is the `MambaLMHeadModel`, which is a Mamba backbone mixer model with a language modeling head. Both Mamba and Mamba-2 share the same overall architecture. They differ only in the mixer component. The overall structure of the Mamba model and its components is illustrated in Figure 13. The main difference between Mamba and Mamba-2's neural network architecture is that the SSM parameters, $A$, $B$, and $C$ in Mamba-2 are produced in parallel with the $X$ input instead of sequentially as in Mamba (Dao & Gu, 2024). The input is first

turned into embeddings and then passes through a varying number of blocks (N) containing Mamba blocks before reaching the final language modeling head for the final output.

**Mamba** is a sequence model that achieves high modeling quality on dense modalities while addressing transformers' inefficiency in handling long sequences (Gu & Dao, 2024). It builds on the S4 model, a type of SSMs, which employs time-invariant dynamics and performs well on continuous data such as audio. SSMs have been less effective with discrete data types because their fixed, uniform processing does not account for how each token's importance at a certain step can change depending on the surrounding tokens in a sequence (Gu & Dao, 2024); for example, like in text or SMILES strings, related tokens may be far apart in a sequence, creating long-range dependencies that are more challenging to capture. To address this limitation, Mamba has introduced selective SSMs (S6), which dynamically adjust parameters ($\Delta$, $B$, and $C$) based on input (see Figure 11). This time-variant nature allows better content-aware processing. Yet, it complicates convolution-like operations and parallelization. To overcome this, Mamba adopts a parallel scan algorithm and hardware-aware optimizations like kernel fusion and recomputation to greatly improve efficiency on GPUs. Unlike transformers, which store full sequence context through key-value caches, making them computationally heavy, Mamba filters and propagates information more efficiently with input-dependent recurrence (Gu & Dao, 2024; Vaswani et al., 2017). Mamba's design enables faster training and inference on longer sequences than transformers, while achieving lower perplexity and strong performance, especially in bioinformatic and cheminformatic tasks (Gu & Dao, 2024; Brazil et al., 2024; Xu et al., 2024; Thoutam & Ellsworth, 2024; Sgarbossa et al., 2024; Peng et al., 2024).

**Mamba-2** builds on Mamba's architecture with a structured state space duality (SSD) framework, showing that SSMs and causal linear attention can be represented as semi-separable matrices (Dao & Gu, 2024). This allows for efficient intra- and inter-chunk computations. By incorporating optimization techniques like tensor and sequence parallelism, Mamba-2 has been reported to further improve hardware efficiency, achieving 2 to 8 times faster training than Mamba, while maintaining transformer-level performance (Dao & Gu, 2024). However, it simplifies the SSM parameter $A$ to a scalar-times-identity form and generates parameters $A$, $B$, and $C$ in parallel from input $X$ (instead of sequentially like in Mamba (see Figure 12), reducing contextual sensitivity and potentially limiting model expressiveness (Dao, 2024).

Figure 11: **Selective SSM (S6)** maps an input $x$ to an output $y$ by transitioning through a higher-dimensional latent state $h$, utilizing time-variant, input-dependent parameters $\Delta$, $B$, and $C$ ($A$ being affected indirectly via discretization with $\Delta$) with an algorithm optimized for efficient use of GPU memory hierarchy levels (Gu & Dao, 2024).

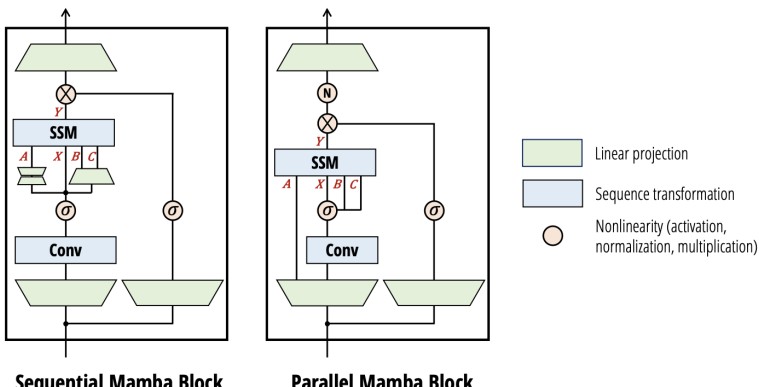

Figure 12: **Sequential and parallel mamba blocks.** The Mamba-2 block (right) simplifies the original Mamba block (left) by generating SSM parameters ($A$, $B$, and $C$) directly from the input $X$ in parallel at the start of the block instead of computing them sequentially as functions of the input $X$, thereby making it more suitable for scaling method such as tensor parallelism (Dao & Gu, 2024; Dao, 2024).

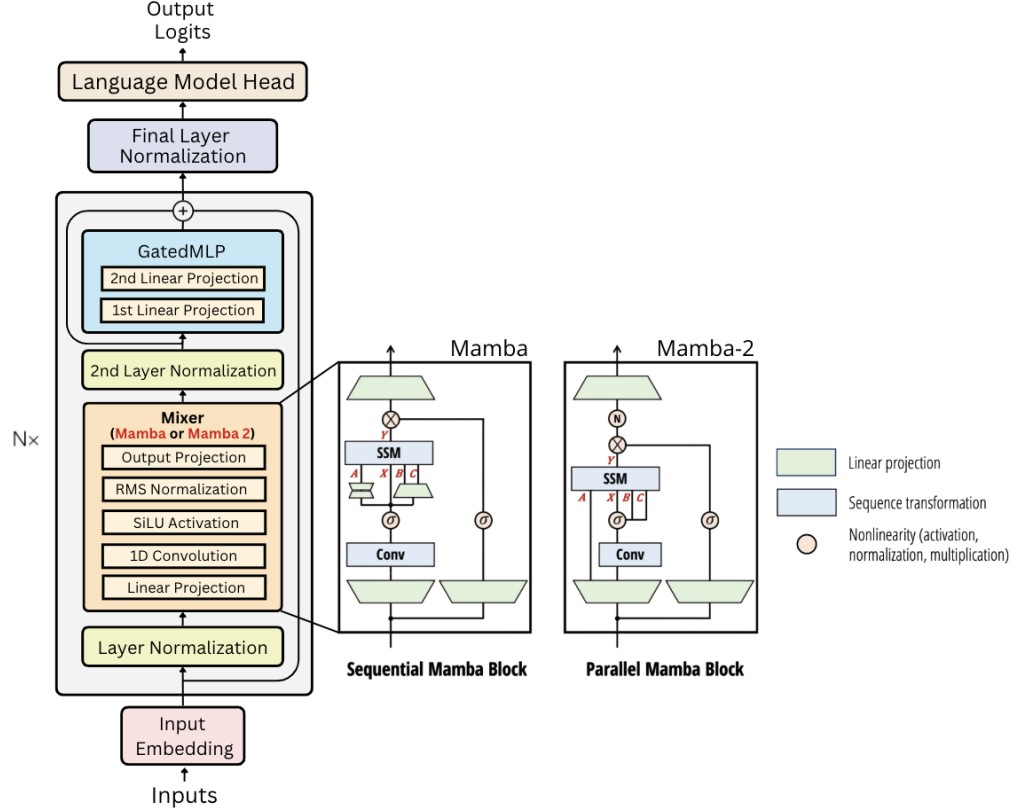

Figure 13: Mamba language model architecture (adapted from Gu & Dao (2024); Dao & Gu (2024)).

### A.4.3 MOLFORMER

MoLFormer is an encoder-only transformer-based CLM designed to capture chemical representations from SMILES sequences using rotary positional embeddings, a linear attention mecha-

Table 6: Comparison of Mamba and Mamba-2 (Dao, 2024).

| Feature | Mamba | Mamba-2 |
|---|---|---|
| SSM Design | Diagonal structure, independent channel control. | Scalar-times-identity structure, uniform dynamics. |
| Hardware Efficiency | Less optimal matrix utilization. | Optimized matrix multiplications for GPUs/TPUs. |
| Theoretical Framework | No structured duality framework. | Introduces SSD, linking SSMs and attention mechanisms. |
| Head Dimension | Single head per channel. | Larger head dimensions (e.g., 64), shared dynamics across channels. |
| State Dimension | Smaller (e.g., N=16), limited scalability. | Larger (e.g., N=64 to N=256), enhances scalability and training speed. |
| Efficiency & Scalability | Flexible control, less training efficiency. | Highly efficient in training, streamlined structure. |
| Training vs. Inference | Assumed better inference performance (more expressive) due to flexibility. | Fast training optimized (sequences longer than 2K), maintains efficient inference. |

nism, and Regex-based tokenization (Ross et al., 2022). The `MoLFormer-XL` has been pre-trained on 1.1 billion unlabelled molecules from the PubChem and ZINC databases. The PubChem database includes a wide range of chemical substances, from small molecules to larger macro-molecules, while the ZINC database focuses on offering commercially available compounds curated for virtual screening. The MoLFormer model utilized in this study is one of its variants, the `MoLFormer-XL-both-10pct`, which has been trained on a 10% subset from both the PubChem and ZINC datasets. Results indicate that the `MoLFormer-XL-both-10pct` variant performed very close and only slightly inferior to the full-scale `MoLFormer-XL`, which has been trained on a dataset ten times larger.

**Fine-tuning MoLFormer** on the random split 1M NP dataset uses the same MLM pre-training objective. The 1M NP dataset is tokenized using the same tokenizer from MoLFormer, resulting in a total of 149 tokens. To identify optimal hyperparameters, 5% of the training and validation data were used for hyperparameter tuning with Optuna, testing learning rates between 1e-5 and 5e-5 and batch sizes of 8, 16, and 32 over 10 trials. The best-performing configuration was then used for full fine-tuning, employing weight decay with early stopping based on validation loss, with a patience value of 5 epochs to prevent overfitting. The final model was evaluated on a held-out test set. After fine-tuning on 1M NPs, the MoLFormer model is adapted for classification by adding a dropout layer (dropout rate = 0.1) and a randomly initialized fully connected linear layer as the classification head. The sequence representations from the last hidden state are extracted as a summary of the entire sequence that is later fed into a classification head to learn task-specific features and generate predictive outputs.

### A.4.4 CHEMBERTA-2

ChemBERTa-2 is a RoBERTa-based transformer model built upon ChemBERTa (Ahmad et al., 2022). Its pre-training dataset consists of 77 million unique SMILES strings sourced from Pub-Chem, which includes a diverse range of small molecules as well as larger natural and chemically modified compounds such as nucleotides, carbohydrates, lipids, and peptides. The multi-task regression (MTR) version of ChemBERTa-2 generally outperforms the MLM version on downstream tasks. The MLM variant has been chosen for fine-tuning on 1M NPs in this case because it is less computationally expensive and time-consuming, as the MTR variant requires extracting 200 molecular features per data point, leading to large feature vectors that substantially increase training time.

The MTR variant is, however, deployed for the downstream property prediction task directly without being fine-tuned on 1M NPs.

One important limitation worth mentioning is that ChemBERTa models have several tokenizer-related issues. A particularly relevant one in this case is that it cannot distinguish chiral centers, yet they are prevalent in NPs. When encountering chiral notations like [C@@H] and [C@H], the ChemBERTa-2 tokenizer tokenizes them into simply "C," effectively losing stereochemical information. Due to this limitation, applying ChemBERTa-2's tokenizer to the 1M NP dataset produces only 32 unique tokens, which is even less than the vocabulary size of the character-level tokenizer in this study. This can impact the model's ability to differentiate structural features. Further discussion on this topic can be found in Section A.6.5.

**Fine-tuning ChemBERTa-2** (ChemBERTa-77M-MLM) adopts the same pre-training MLM objective, and the fine-tuning process is the same as that of MoLFormer. The 1M NP dataset is tokenized with the original ChemBERTa-2's tokenizer. The hyperparameter search uses 5% of the training and validation data with Optuna, with a search space of learning rates between 1e-5 and 5e-5 and batch sizes of 8, 16, and 32 over 10 trials. The best configuration was used for full fine-tuning with weight decay and early stopping (patience = 5 epochs) based on validation loss. After ChemBERTa-77M-MLM is fine-tuned on 1M NPs, it is then adapted for property prediction tasks by adding a classification head consisting of a dropout layer (dropout rate = 0.1) and a randomly initialized fully connected linear layer, the same as all other models. The forward pass extracts the contextualized representation of each sequence, namely, using the [CLS] token embedding, which is the first token embedding from the last hidden state. During fine-tuning for property prediction tasks, the model learns to encode task-relevant features into this embedding, making it a meaningful sequence summary representation for classification.

## A.5 Related Work (Extended)

**NIMO**, developed by Shen et al. (2024), is a transformer-based model designed for NP-inspired molecular generation. It employs a motif-based representation with custom fragmentation methods to preserve stereochemical information and enhance structural design. Trained on datasets that are mostly NPs, including COCONUT, ChEMBL, and TeroKIT, NIMO has shown success in generating diverse, drug-like molecules with high structural relevance. **NPGPT** is developed by Sakano et al. (2024) for NP-like molecular generation. The authors fine-tuned SMILES-GPT (GPT-2-based) and ChemGPT (GPT-Neo-based) models on around 400,000 NPs from the COCONUT database, which was then expanded to around 3.6 million molecules through augmentation. The augmentation process involved enumerating multiple valid SMILES representations for the same molecule by altering the traversal order of atoms during encoding. The fine-tuned models generated molecules that closely matched real NPs in terms of molecular weight, stereochemistry, and NP-likeness. The study also demonstrates how deep learning can accelerate NP-based drug discovery by efficiently navigating complex chemical spaces. Feller & Wilke (2025) have introduced **PeptideCLM**, a transformer-based CLM designed to predict the membrane permeability of cyclic peptides, a subset of NPs with significant pharmaceutical potential. The authors used a pre-training dataset of 23 million molecules, including 10 million small molecules from PubChem, another 2.2 million small molecules from SureChEMBL, 825,632 natural peptides, and 10 million synthetic peptides. The models reportedly address limitations in peptide representation using a custom tokenization strategy based on SMILES Pair Encoding (SPE) within a BERT-style architecture, and they have been successfully applied to predict peptide permeability. Tay et al. (2023) trained a recurrent neural network (RNN) with long short-term memory (LSTM) units using the COCONUT database, which contains approximately 406,000 NP molecules. They applied SMILES enumeration to augment the training dataset, expanding it to around 3.25 million entries, following a similar approach to what Sakano and Furui have adopted. Using this model, they generated 100 million molecules, of which around 67 million were unique and valid. Their findings indicate that the generated compounds expand the known chemical space while preserving NP characteristics, closely resembling real NPs in NP-likeness scores and biosynthetic pathway classification, with low Kullback-Leibler (KL) divergence. Özçelik et al. (2024) have used structured state-space sequence (S4) model, GPT, and LSTM to generate NP-like molecules, training on 32,360 data points from the COCONUT database to produce 102,400 NP-like SMILES strings de novo. They have shown that the S4 model, which is the predecessor to Mamba, can enhance the generation of NP-like molecules by generating higher percentages of diverse and structurally novel molecules compared to GPT, RNN, and LSTM.

Recent research in CLMs for NPs has demonstrated that LLMs can effectively expand the chemical space of NPs by generating NP-like molecules. However, to the best of our knowledge, no studies have investigated the use of Mamba models for NP-focused CLMs, nor have any systematically examined tokenization strategies for NPs on property prediction tasks. Furthermore, some models are pre-trained on molecular datasets that cover a broader chemical space before being fine-tuned for NPs. This prompts the question of whether it is more effective to pre-train models solely on NPs or to fine-tune existing CLMs using NP data. Therefore, this work aims to address these gaps.

## A.6 TOKENIZATION

Tokenization converts molecular representations into tokens for processing by language models. SMILES tokenization differs from natural languages. Natural language tokens represent words or subwords and must handle punctuation, ambiguity, and more redundancy that come with the use of stop words. Chemical text representations like the SMILES strings, however, are designed to encode atoms, bonds, branches, and rings in a more compact way. SMILES tokenization can be character-, atom-, or motif-level (Liao et al., 2024). Character-level tokenizer treats each character as a token. Atom-level focuses on individual atoms. Motif-level uses substructures, derived either from chemical rules or subword algorithms like byte-pair encoding (BPE). While motif- and atom-level methods are more chemically grounded, character-level tokenization has also shown to be effective (Lu & Zhang, 2022). Some studies have shown that optimized tokenizers such as BPE, WordPiece, and Unigram can enhance transformer-based CLM performance while reducing input length (Dotan et al., 2024). SMILES Pair Encoding (SPE), a BPE variant, improves generation and prediction by incorporating chemical meaning (Li & Fourches, 2020). Atom-in-SMILES (AIS) includes atomic environments, outperforming character-level and SPE in property prediction tasks (Ucak et al., 2023).

### A.6.1 CHARACTER-LEVEL TOKENIZER

Character-level tokenization is a straightforward method that divides SMILES strings into individual characters, sometimes leading to separation of multi-character units. However, research has shown its effectiveness in some CLMs (Liao et al., 2024). Tokenizing the 1M NP dataset by the character-level tokenizer results in a vocabulary size of 48, including four special tokens (padding, unknown, beginning, and end of sequence tokens).

### A.6.2 DEEPCHEM BYTE-PAIR ENCODING (BPE) TOKENIZER

The tokenizer sourced from DeepChem (`seyonec/PubChem10M_SMILES_BPE_450k` [18]) is a BPE tokenizer with a vocabulary size of 7,924 tokens. In this study, this tokenizer is simply referred to as "BPE." This tokenizer is not specifically tailored for NPs. Therefore, when applied to 1M NPs, only about 1,700 out of the total 7,924 tokens are included, and much of the tokenizer's broader vocabulary remains unused.

### A.6.3 ATOM-IN-SMILES (AIS) TOKENIZER

The atom-level tokenizer adopted in this study is the Atom-in-SMILES (AIS) tokenizer developed by Ucak et al. (2023). It is a tokenization scheme designed to address limitations in traditional SMILES tokenization by including an atom's local chemical context or atom environments (AEs) in its token. AIS tokenization can also mitigate token degeneration issues by reducing token repetition. The AIS algorithm processes the input strings by first parsing the SMILES string into a chemical graph, identifying atoms and bonds. It then analyzes each atom's local chemical environment by detecting neighboring atoms within a specific bond radius, examining bond types, and capturing features like aromaticity, chirality, and hybridization. Based on this analysis, AIS generates context-aware tokens for each atom, incorporating the central atom, neighboring atoms, bond types, and special chemical properties. For example, the AIS tokenization output of ethanol's SMILES string "CCO" are "[CH3;!R;C], [CH2;!R;CO], [OH;!R;C]," where [CH3;!R;C] denotes a methyl group not in a ring, bonded to a carbon; [CH2;!R;CO] is a methylene group not in a ring, bonded to both a carbon and an oxygen; [OH;!R;C] represents a hydroxyl group not in a ring, bonded to a carbon.

---

[18] https://huggingface.co/seyonec/PubChem10M_SMILES_BPE_450k

Tokenizing the 1M NP pre-training dataset by the AIS tokenizer results in a vocabulary size of 1023, including the special tokens. It is also important to note that the AIS tokenizer may occasionally interchange the counterclockwise ("@") and clockwise ("@@") chiral designations when mapping SMILES to AIS tokens. This phenomenon arises from RDKit's internal processing of chiral centers, where chirality is assigned based on the atom's local environment.

### A.6.4 NATURAL PRODUCT BYTE-PAIR ENCODING (NPBPE) TOKENIZER

The Natural Product Byte-Pair Encoding (NPBPE) tokenizers are made specifically for NPs. They are trained using the `tokenizers.trainers.BpeTrainer` class from Hugging Face on the 1M NP dataset. The process involves using a pre-tokenizer that isolates individual SMILES characters and then iteratively merges frequent character pairs to create more extended units. Five distinct NPBPE tokenizers are created by setting maximum vocabulary sizes to 60, 100, 1000, 7924, and 30,000 tokens (including special tokens).

Table 7: Examples of tokenized SMILES strings for each tokenizer.

Example SMILES: `CC1=C2[C@@H]3[C@H](C(=O)C1)[C@@]2(C)CCCC3(C)C`

| Tokenizer | Tokens |
|---|---|
| Character-level | C, C, 1, =, C, 2, [, C, @, @, H, ], 3, [, C, @, H, ], (, C, (, =, O, ), C, 1, ), [, C, @, @, ], 2, (, C, ), C, C, C, C, 3, (, C, ), C |
| AIS | [CH3;!R;C], [C;R;CCC], 1, =, [C;R;CCC], 2, [[C@H];R;CCC], 3, [[C@H];R;CCC], (, [C;R;CCO], (, =, [O;!R;C], ), [CH2;R;CC], 1, ), [[C@@];R;CCCC], 2, (, [CH3;!R;C], ), [CH2;R;CC], [CH2;R;CC], [C;R;CCCC], 3, (, [CH3;!R;C], ), [CH3;!R;C] |
| BPE | CC, 1, =, C, 2, [, C, @, @, H, ], 3, [, C, @, H, ](, C, (=, O, ), C, 1, )[, C, @, @, ], 2, (, C, ), CCCC, 3, (, C, ), C |
| NPBPE60 | CC, 1, =, C, 2, [C@, @, H, ], 3, [C@, H, ], (, C(, =, O), C, 1, ), [C@, @, ], 2, (, C, ), CC, CC, 3, (, C, ), C |
| NPBPE100 | CC1, =, C2, [C@@H], 3, [C@H](, C(=O), C1, ), [C@@], 2, (C), CCCC, 3, (C), C |
| NPBPE1000 | CC1=C2, [C@@H]3, [C@H](, C(=O), C1), [C@@]2(C), CCCC, 3(C), C |
| NPBPE7924 | CC1=C2, [C@@H]3[C@H](, C(=O), C1), [C@@]2(C), CCCC3(C), C |
| NPBPE30k | CC1=C2, [C@@H]3[C@H](, C(=O), C1), [C@@]2(C), CCCC3(C)C |

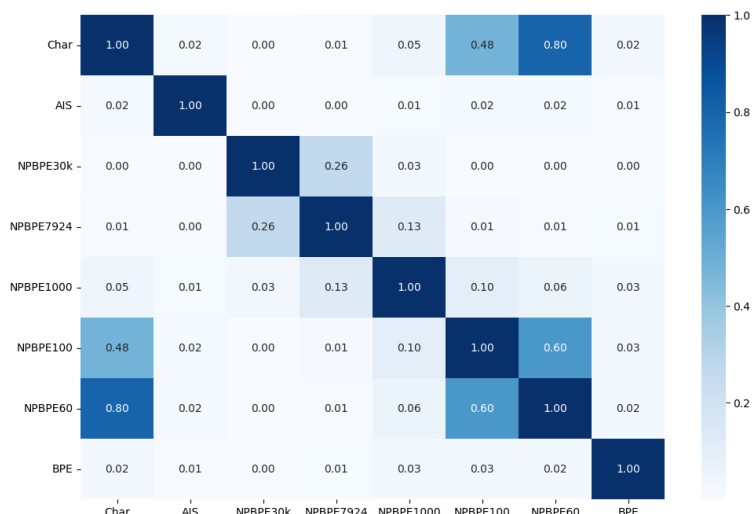

Figure 14: **Jaccard similarity between tokenizers.** The Jaccard similarity is calculated by dividing the size of each tokenizer pair's token intersection by the size of their union; for BPE, only the tokens actually used in 1M NPs (approximately 1,700) are included.

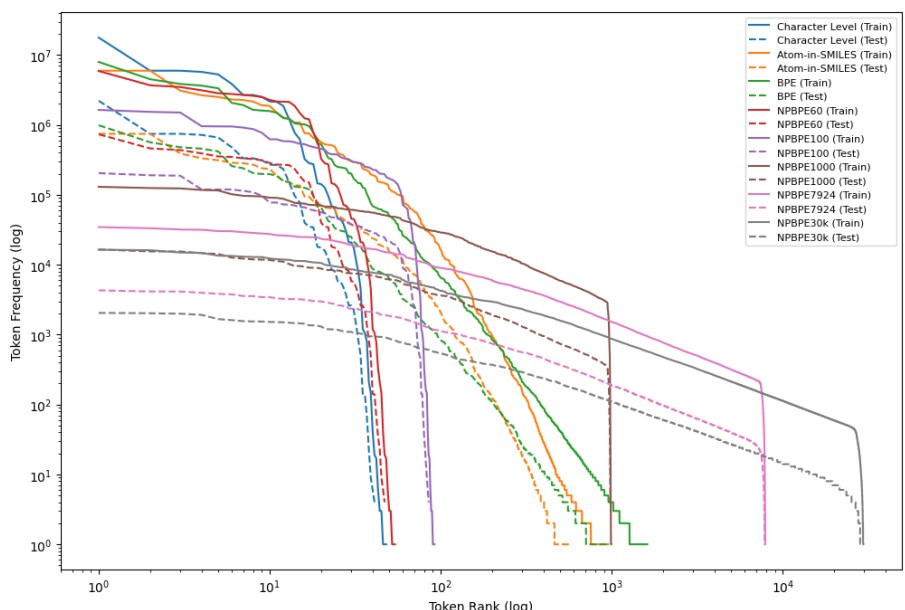

Figure 15: **Zipfian distributions of tokens generated by each tokenizer.** The training and testing sets shown in the plot are the scaffold split subsets of the 1M NP dataset.

### A.6.5    WHEN TOKENIZATION FAILS....

MoLFormer's fine-tuning on 1M NPs took approximately 10 days, whereas ChemBERTa-77M-MLM completed fine-tuning in about 12 hours. Although differences in architecture and model size likely contribute to this discrepancy, another possible factor is the diversity of their tokenizer outputs: ChemBERTa-2's tokenizer generated only 32 distinct tokens from the 1M NP dataset, in contrast to MoLFormer's 149. This disparity results from known tokenization issues in ChemBERTa models—for example, stereochemical details are simplified to basic tokens, and bracketed annotations and charge indicators are omitted—causing distinct SMILES components to become identical tokens. Although the impact of this tokenizer issue on training and inference performance has not

been formally assessed, it can already be observed that ChemBERTa-77M-MLM, both its fine-tuned and non-fine-tuned versions, consistently underperform compared to all other models in this study. This result aligns with the ChemBERTa-2 paper, which also reports the superior performance of the multi-task regression (MTR) variant over MLM. It should be noted that ChemBERTa-2's MTR pre-trained variant also has the same tokenization issue. However, despite this issue, it still performs reasonably well compared to other models without tokenizer issues. One can perhaps speculate that the MTR pre-training objective may be inherently more resilient to poor tokenization than MLM. While MLM relies heavily on token-level relationships, requiring precise tokenization to reconstruct masked tokens, MTR leverages 200 precomputed molecular features calculated directly from SMILES strings, and the model has been trained to predict all 200 properties simultaneously (Ahmad et al., 2022). These numerical features, derived using RDKit, provide chemical contexts that allow the model to focus on molecular features and not rely solely on token-level granularity. As a result, MTR pre-training can perhaps be less dependent on the quality of tokenized input and more driven by the underlying chemical patterns and relationships embedded in numerical descriptors, thereby enabling the model to learn more robust, chemically grounded representations even when tokenized input is suboptimal. Nonetheless, this remains a speculation, and further research is necessary to determine the exact mechanisms behind this robustness and whether fixing the tokenizer could enhance the MLM variant's performance.

## A.7 FINE-TUNING ON DOWNSTREAM TASKS

**Hyperparameter optimization** is performed using only the training and validation subsets. Each fold undergoes a full fine-tuning process for three epochs, and the evaluation metric is recorded after the third epoch. The evaluation results for all 5 folds are then averaged. This process is repeated for each of the six hyperparameter combinations in the search space (learning rates: 1e-4, 1e-5, 5e-5; batch sizes: 8, 16). The combination resulting in the highest average performance across the 5 folds on the validation data is selected for fine-tuning. Figure 16 illustrates the hyperparameter search process for all property prediction tasks in this study.

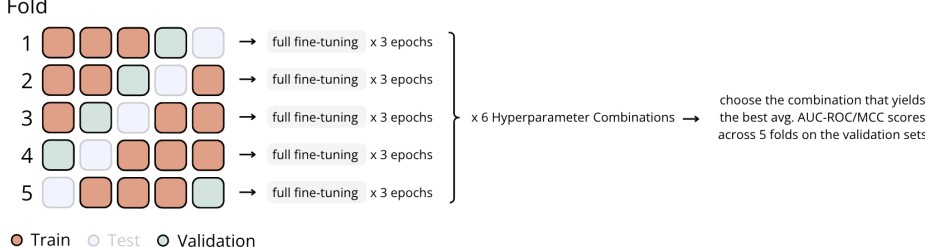

Figure 16: 5×5-fold cross-validation - hyperparameter search.

**Fine-tuning** on property prediction tasks is performed using full fine-tuning on each fold, with early stopping applied with the patience value set to 5 epochs. This fine-tuning process is repeated 5 times per fold, with each repetition starting from the original model weights (i.e., prior to any modifications using the downstream task data). The test results from these 5 repetitions per fold are averaged to obtain a more reliable performance estimate. After all 5 folds have been processed, the final results are computed by averaging the respective fold-wise outcomes, along with the corresponding standard deviation and standard error (SE). This repeated 5×5-fold cross-validation fine-tuning process is illustrated in Figure 17.

**Loss functions** for each task differ slightly. For the FourTastes dataset, `CrossEntropyLoss()` is used for multi-class classification, as the task involves four taste categories. This loss function applies a softmax activation to convert logits into class probabilities. Due to the highly imbalanced label distribution, class weighting is applied to mitigate class imbalance. The class weights are computed using `compute_class_weight()` from `sklearn` and incorporated into the loss function, ensuring higher loss penalties for underrepresented classes. For the anti-cancer activity and peptide permeability datasets, class distributions are approximately balanced; thus, no class weighting is applied. These two binary classification tasks use the `BCEWithLogitsLoss()` function, which combines a sigmoid activation with binary cross-entropy loss.

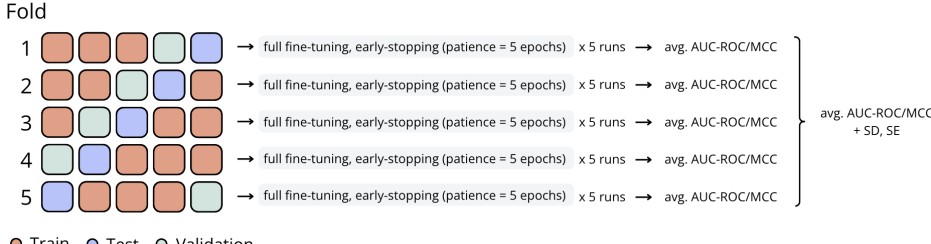

Figure 17: 5×5-fold cross-validation - fine-tuning.

Table 8: **Model performance comparison** for anti-cancer activity (MCC±SE) and peptide permeability (AUC-ROC±SE). This table presents the same results as those shown in Figure 4, but rendered in tabular form for clarity.

| | | Fine-tuned on 1M NPs | | Pre-trained on 1M NPs |
|---|---|---|---|---|
| | | ChemBERTa-77M-MLM | MoLFormer-XL-both-10pct | NP Mamba (M1-AIS-rds) |
| Anti-Cancer Activity | Random | 0.779±0.004 | 0.810±0.003 | **0.814±0.003** |
| | Scaffold | 0.638±0.023 | 0.681±0.021 | 0.673±0.020 |
| Peptide Permeability | Random | 0.847±0.005 | 0.855±0.006 | **0.863±0.004** |
| | Scaffold | 0.755±0.018 | 0.759±0.027 | **0.777±0.013** |
| | | Without Fine-tuning on 1M NPs | | |
| | | ChemBERTa-77M-MLM | MoLFormer-XL-both-10pct | ChemBERTa-77M-MTR |
| Anti-Cancer Activity | Random | 0.777±0.001 | **0.814±0.002** | 0.806±0.003 |
| | Scaffold | 0.633±0.019 | **0.689±0.021** | 0.680±0.026 |
| Peptide Permeability | Random | 0.845±0.005 | 0.855±0.005 | **0.863±0.004** |
| | Scaffold | 0.751±0.018 | 0.732±0.031 | 0.775±0.018 |

Figure 18 presents the multiple comparisons similarity (MCSim) plots for the FourTastes results under both random and scaffold splits using random split pre-trained models, providing easier pairwise comparisons and highlighting statistically significant differences. The MCSim plots show that differences between tokenizers are more pronounced than between models, and tokenizers using larger NPBPE vocabularies perform significantly worse than most other tokenizers.

### A.8 MOLECULE GENERATION DETAILS

Molecule generation is conducted using an autoregressive sampling approach with a temperature of one, which means the model samples directly from its learned probability distribution. Each of the 48 model variations generates 100,000 molecules in a batch-wise manner, with a maximum sequence length of 512 tokens and a batch size of 32. Generation begins with a start-of-sequence token and proceeds token-by-token, sampling from the model's probability distribution until the end-of-sequence token is reached. Generation time is recorded for performance assessment. The generated molecules are evaluated for validity, uniqueness, and novelty, with their NP-likeness and SAScores compared to their respective training data (see Section A.11 for details regarding NP-likeness and SAScores). More information on molecule generation results can be found in Figures 19 and 20 below.

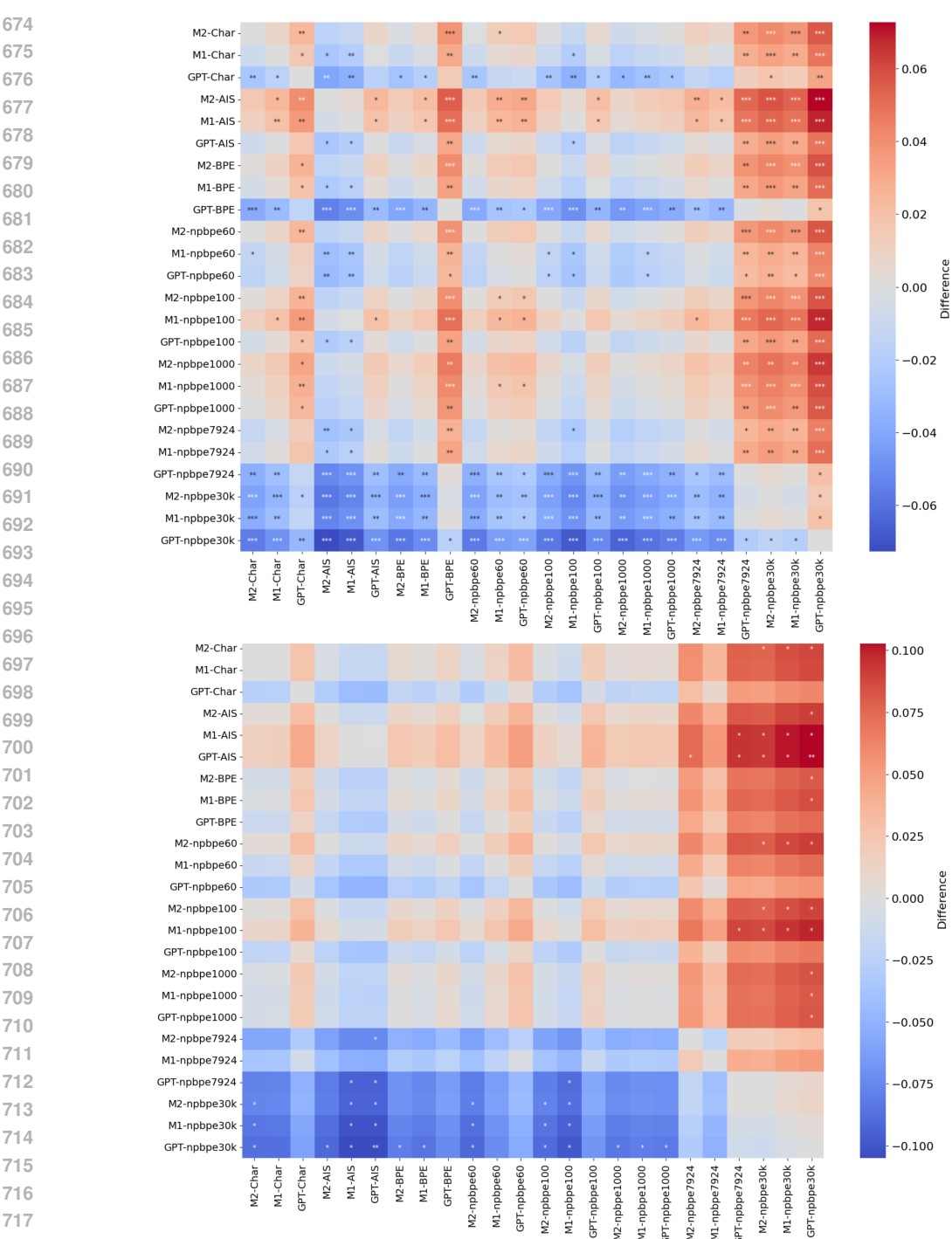

Figure 18: **Multiple comparisons similarity (MCSim) - random (top) and scaffold (bottom) split FourTastes + random split pre-trained models**: this heatmap visualizes pairwise differences in model performance, with statistical significance indicated by asterisks (*); pairwise comparisons are conducted using Welch's t-test, which compares means while accounting for unequal variances; degrees of freedom are estimated using the Welch-Satterthwaite equation, and two-tailed p-values are calculated using the Student's t-distribution; statistical significance is assigned based on p-values: $***$ $(p < 0.001)$, $**$ $(p < 0.01)$, and $*$ $(p < 0.05)$.

| Metric | Model | Character-Level | Atom-in-SMILES | BPE | NPBPE60 | NPBPE100 | NPBPE1000 | NPBPE7924 | NPBPE30k |
|---|---|---|---|---|---|---|---|---|---|
| Tokenizer Vocabulary Size | | 48 | 1023 | 7924 | 60 | 100 | 1000 | 7924 | 30000 |
| Averaged Tokenized Sequence Length | | 74.9 | 62.97 | 58.48 | 32.4 | 32.4 | 22.45 | 10.9 | 8.57 |
| Valid | M2 | 80.93% | 81.57% | 73.68% | 82.03% | 76.58% | 80.65% | 65.17% | 56.98% |
| | M1 | 81.21% | 81.89% | 80.33% | 79.07% | 85.36% | 76.96% | 75.86% | 57.33% |
| | GPT | 79.91% | 80.82% | 80.25% | 79.89% | 79.86% | 84.83% | 64.46% | 57.55% |
| Unique | M2 | 80.09% | 80.49% | 72.85% | 81.00% | 75.59% | 78.97% | 63.55% | 54.83% |
| | M1 | 80.34% | 80.76% | 79.39% | 78.16% | 84.28% | 75.58% | 73.61% | 55.13% |
| | GPT | 79.14% | 79.50% | 79.12% | 78.85% | 78.50% | 82.86% | 62.02% | 54.87% |
| Novel (train/all data) | M2 | 70.12 / 67.94% | 71.31 / 69.18% | 64.17 / 62.22% | 70.00 / 67.61% | 69.65 / 68.20% | 62.32 / 59.52% | 56.09 / 54.63% | 46.54 / 45.11% |
| | M1 | 70.86 / 68.73% | 71.42 / 69.36% | 69.88 / 67.78% | 68.34 / 66.07% | 71.11 / 68.34% | 68.88 / 67.36% | 54.30 / 51.41% | 47.25 / 45.79% |
| | GPT | 75.09 / 74.01% | 74.03 / 72.70% | 71.72 / 69.99% | 72.21 / 70.62% | 69.13 / 67.01% | 69.00 / 66.01% | 55.03 / 53.50% | 46.39 / 44.82% |
| NP-Likeness | M2 | 1.1088 ± 0.0046 | 1.1300 ± 0.0046 | 1.0695 ± 0.0047 | 1.1180 ± 0.0047 | 1.0898 ± 0.0046 | 1.1329 ± 0.0047 | 1.0770 ± 0.0049 | 0.9964 ± 0.0054 |
| | M1 | 1.1204 ± 0.0046 | 1.1162 ± 0.0046 | 1.0745 ± 0.0046 | 1.0851 ± 0.0047 | 1.1145 ± 0.0045 | 1.0817 ± 0.0046 | 1.1053 ± 0.0049 | 1.0166 ± 0.0053 |
| | GPT | 1.0099 ± 0.0047 | 1.0651 ± 0.0046 | 0.9845 ± 0.0047 | 0.9773 ± 0.0046 | 1.1371 ± 0.0047 | 1.1138 ± 0.0047 | 0.9482 ± 0.0052 | 0.9930 ± 0.0054 |
| SAScore | M2 | 4.2507 ± 0.0045 | 4.2398 ± 0.0046 | 4.1795 ± 0.0046 | 4.2305 ± 0.0045 | 4.2607 ± 0.0047 | 4.2669 ± 0.0046 | 4.2464 ± 0.0049 | 4.1652 ± 0.0053 |
| | M1 | 4.2822 ± 0.0045 | 4.1981 ± 0.0045 | 4.2046 ± 0.0044 | 4.2529 ± 0.0046 | 4.2984 ± 0.0046 | 4.3246 ± 0.0047 | 4.2625 ± 0.0047 | 4.1736 ± 0.0053 |
| | GPT | 4.2145 ± 0.0044 | 4.2218 ± 0.0045 | 4.2233 ± 0.0044 | 4.2337 ± 0.0044 | 4.3527 ± 0.0046 | 4.3195 ± 0.0045 | 4.1050 ± 0.0048 | 4.1141 ± 0.0051 |
| Generation Time (per molecule) | M2 | 0.2756 sec | 0.1493 sec | 0.0707 sec | 0.1536 sec | 0.0451 sec | 0.0294 sec | 0.0142 sec | 0.0094 sec |
| | M1 | 0.1803 sec | 0.1257 sec | 0.1893 sec | 0.0482 sec | 0.0522 sec | 0.0141 sec | 0.0084 sec | 0.0062 sec |
| | GPT | 0.1658 sec | 0.0410 sec | 0.1201 sec | 0.0453 sec | 0.0167 sec | 0.0050 sec | 0.0097 sec | 0.0055 sec |
| Generation Time (total) | M2 | 7 hr 39 min | 4 hr 9 min | 1 hr 59 min | 4 hr 16 min | 1 hr 15 min | 45 min | 23 min | 15 min |
| | M1 | 5 hr | 3 h 29 min | 5 hr 17 min | 1 hr 20 min | 1 hr 27 min | 23 min | 14 min | 10 min |
| | GPT | 4 hr 36 min | 1 hr 8 min | 3 hr 22 min | 1 hr 15 min | 27 min | 8 min | 16 min | 9 min |
| Unique Scaffolds | M2 | 42676 | 40497 | 39165 | 42023 | 43775 | 38908 | 36620 | 30842 |
| | M1 | 43251 | 40844 | 41588 | 41374 | 44101 | 44929 | 35729 | 31027 |
| | GPT | 44996 | 38701 | 43364 | 44513 | 40600 | 40317 | 34151 | 28680 |
| Novel Scaffolds (v.s. training data) | M2 | 31296 | 28393 | 29783 | 29838 | 36290 | 24395 | 29493 | 23820 |
| | M1 | 32444 | 29572 | 31182 | 30295 | 30340 | 37144 | 20885 | 24079 |
| | GPT | 37821 | 29898 | 35212 | 36184 | 30710 | 27148 | 27719 | 21723 |
| Novel Scaffolds (v.s. all data) | M2 | 31194 | 28279 | 29692 | 29724 | 36189 | 24300 | 29415 | 23746 |
| | M1 | 32339 | 29463 | 31097 | 30192 | 30217 | 37051 | 20799 | 23996 |
| | GPT | 37740 | 29810 | 35128 | 36087 | 30609 | 27034 | 27658 | 21659 |

Figure 19: Molecule generation result table (random split pre-trained models).

| Metric | Model | Character-Level | Atom-in-SMILES | BPE | NPBPE60 | NPBPE100 | NPBPE1000 | NPBPE7924 | NPBPE30k |
|---|---|---|---|---|---|---|---|---|---|
| Tokenizer Vocabulary Size | | 48 | 1023 | 7924 | 60 | 100 | 1000 | 7924 | 30000 |
| Averaged Tokenized Sequence Length | | 74.9 | 62.97 | 58.48 | 32.4 | 32.4 | 22.45 | 10.9 | 8.57 |
| Valid | M2 | **82.87%** | 80.45% | **79.50%** | 74.53% | 79.68% | 78.64% | **72.99%** | 47.40% |
| | M1 | 80.45% | 80.62% | 79.42% | 80.85% | 78.92% | 76.64% | 67.88% | **57.46%** |
| | GPT | 74.45% | **82.21%** | 77.46% | **81.76%** | **84.46%** | **83.67%** | 66.24% | 45.89% |
| Unique | M2 | **81.53%** | 79.01% | **78.27%** | 73.50% | 78.00% | 76.11% | **69.84%** | 45.25% |
| | M1 | 79.35% | 79.30% | 78.12% | 79.77% | 77.74% | 74.74% | 65.52% | **54.55%** |
| | GPT | 73.62% | **80.89%** | 76.27% | **80.63%** | **82.61%** | **81.23%** | 63.38% | 42.48% |
| Novel (train/all data) | M2 | 66.63 / 66.45% | 66.63 / 66.50% | 64.73 / 64.61% | 66.08 / 66.03% | 68.10 / 68.02% | 56.14 / 55.96% | 46.49 / 46.34% | 37.39 / 37.32% |
| | M1 | 66.20 / 66.03% | 67.93 / 67.80% | 64.94 / 64.80% | 66.81 / 66.65% | 64.23 / 64.08% | **64.49 / 64.39%** | **54.52 / 54.40%** | **42.77 / 42.68%** |
| | GPT | **68.40 / 68.34%** | **72.37 / 72.26%** | **66.63 / 66.52%** | **71.50 / 71.40%** | **69.24 / 69.08%** | 63.02 / 62.83% | 52.76 / 52.67% | 33.85 / 33.77% |
| NP-Likeness | M2 | **1.1485** ± 0.0045 | 1.1259 ± 0.0047 | 1.1277 ± 0.0045 | 1.0861 ± 0.0046 | 1.1449 ± 0.0044 | 1.1677 ± 0.0047 | **1.1389** ± 0.0049 | 1.0382 ± 0.0055 |
| | M1 | 1.1403 ± 0.0046 | **1.1819** ± 0.0046 | 1.1592 ± 0.0046 | **1.1730** ± 0.0045 | **1.1734** ± 0.0046 | 1.1095 ± 0.0045 | 1.1372 ± 0.0048 | **1.0667** ± 0.0053 |
| | GPT | 0.9960 ± 0.0047 | 1.0634 ± 0.0047 | **1.1595** ± 0.0047 | 1.0460 ± 0.0045 | 1.1655 ± 0.0045 | **1.1722** ± 0.0047 | 1.0894 ± 0.0050 | 0.9399 ± 0.0057 |
| SAScore | M2 | 4.1387 ± 0.0043 | 4.1309 ± 0.0044 | 4.0906 ± 0.0043 | 4.1219 ± 0.0045 | 4.1341 ± 0.0043 | 4.1753 ± 0.0044 | 4.1697 ± 0.0046 | 4.0715 ± 0.0055 |
| | M1 | **4.1673** ± 0.0043 | 4.1468 ± 0.0043 | 4.1568 ± 0.0043 | 4.2157 ± 0.0044 | 4.1980 ± 0.0045 | 4.1542 ± 0.0045 | **4.2129** ± 0.0048 | **4.1109** ± 0.0051 |
| | GPT | 4.1657 ± 0.0045 | **4.2461** ± 0.0044 | **4.3047** ± 0.0045 | **4.2467** ± 0.0044 | **4.2770** ± 0.0042 | **4.2726** ± 0.0044 | 4.1814 ± 0.0046 | 3.9353 ± 0.0053 |
| Generation Time (per molecule) | M2 | 0.2187 sec | 0.1217 sec | 0.1805 sec | 0.1384 sec | 0.0442 sec | 0.0208 sec | 0.0167 sec | 0.0053 sec |
| | M1 | 0.2302 sec | 0.0801 sec | 0.1745 sec | 0.1394 sec | 0.0455 sec | 0.0110 sec | 0.0045 sec | 0.0070 sec |
| | GPT | 0.1512 sec | 0.0425 sec | 0.1401 sec | 0.0914 sec | 0.0121 sec | 0.0042 sec | 0.0060 sec | 0.0072 sec |
| Generation Time (total) | M2 | 6 hr 4 min | 3 hr 23 min | 5 hr 2 min | 3 hr 50 min | 1 hr 13 min | 34 min | 27 min | 8 min |
| | M1 | 6 hr 23 min | 2 hr 13 min | 4 hr 52 min | 3 hr 52 min | 1 hr 15 min | 18 min | 7 min | 11 min |
| | GPT | 4 hr 12 min | 1 hr 11 min | 3 hr 55 min | 2 hr 32 min | 20 min | 7 min | 10 min | 12 min |
| Unique Scaffolds | M2 | 33199 | 30956 | 32072 | 35252 | **36688** | 28079 | 24884 | 21389 |
| | M1 | 33564 | 31505 | 32655 | 34844 | 32746 | **34947** | **30697** | **24067** |
| | GPT | **35237** | **34458** | **36523** | **38873** | 34945 | 30522 | 28936 | 17457 |
| Novel Scaffolds (v.s. training data) | M2 | 24615 | 22104 | 24149 | 30675 | **31121** | 17993 | 14524 | 17663 |
| | M1 | 25619 | 23178 | 24849 | 26838 | 24396 | **29127** | **24918** | **18540** |
| | GPT | **30607** | **27698** | **30717** | **33048** | 27555 | 20875 | 23954 | 13878 |
| Novel Scaffolds (v.s. all data) | M2 | 22681 | 20156 | 22293 | 29029 | **29374** | 16464 | 13271 | 16856 |
| | M1 | 23707 | 21204 | 22999 | 24963 | 22617 | **27547** | **23583** | **17509** |
| | GPT | **29015** | **25771** | **28933** | **31226** | 25685 | 19171 | 22747 | 13067 |

Figure 20: Molecule generation result table (scaffold split pre-trained models).

## A.9 PSEUDO-NP ERROR ANALYSIS

Long-context modeling is not central to our tasks, as NP SMILES are relatively short (avg.<200 characters); instead, challenges arise from stereochemistry and structural diversity, which increase information density and complicate tokenization, particularly with rare or fused ring systems. To better understand these sources of difficulty in molecule generation, we perform error analysis using `partialsmiles`, a Python library that parses and validates partial SMILES strings by checking syntax, aromatic system kekulization, and atom valences against allowed ranges (O'Boyle, 2020). The parser reports three main types of errors: syntax errors (e.g., illegal characters, unmatched brackets, invalid SMILES structure, and more), valence errors (when an atom's valence is not on the allowed list for its charge state), and kekulization failures (when an aromatic system cannot be resolved into alternating bonds). These errors are raised as exceptions that can be caught programmatically, allowing for the diagnosis of why specific SMILES strings are invalid.

In Table 9, regarding tokenizers' average syntax error, NPBPEs with large vocabularies might have over-merged unrelated symbols that break chemical substructures, thus increasing the percentage of syntax errors. While BPE, NPBPE100, and 1000 seem to provide a more balanced trade-off in granularity and context and reduce the share of syntax errors, AIS still outperforms all tokenizers in validity, producing the highest number of valid molecules.

Furthermore, the error distribution aligns with the Pareto principle (Figure 21 and Table 10), with a few dominant error types (around 20%), such as kekulization failure, and unclosed ring openings, parentheses, and branches, accounting for the majority of failures (around 80%), pointing to key areas for targeted pesudo-NP molecules generation improvement.

Table 9: Model- and tokenizer-wise average error type.

| Error Type | | Char | BPE | AIS | NPBPE60 | NPBPE100 | NPBPE1000 | NPBPE7924 | NPBPE30k | Avg. |
|---|---|---|---|---|---|---|---|---|---|---|
| **Syntax Error** | Mamba-2 | 80.94% | 80.00% | 81.58% | 78.98% | 75.51% | 80.59% | 81.48% | 87.72% | 80.85% |
| | Mamba | 77.07% | 72.10% | 77.47% | 81.11% | 73.85% | 69.35% | 85.22% | 86.61% | 77.85% |
| | GPT | 79.30% | 75.96% | 81.02% | 81.04% | 80.68% | 79.62% | 85.34% | 89.36% | 81.54% |
| | Avg. | 79.10% | 76.02% | 80.02% | 80.38% | 76.68% | 76.52% | 84.01% | 87.90% | |
| **Kekulization Failure** | Mamba-2 | 12.39% | 11.67% | 13.32% | 12.51% | 16.42% | 11.18% | 12.46% | 8.06% | 12.25% |
| | Mamba | 16.18% | 15.37% | 17.52% | 12.40% | 16.79% | 19.82% | 8.93% | 9.01% | 14.50% |
| | GPT | 15.83% | 14.49% | 16.14% | 14.57% | 13.36% | 13.42% | 11.02% | 7.83% | 13.33% |
| | Avg. | 14.80% | 13.84% | 15.66% | 13.16% | 15.52% | 14.81% | 10.80% | 8.30% | |
| **Valence Error** | Mamba-2 | 6.66% | 8.33% | 5.10% | 8.49% | 8.06% | 8.23% | 6.06% | 4.22% | 6.89% |
| | Mamba | 6.74% | 12.52% | 5.01% | 6.48% | 9.35% | 10.83% | 5.85% | 4.38% | 7.65% |
| | GPT | 4.87% | 9.56% | 2.85% | 4.39% | 5.95% | 6.96% | 3.64% | 2.81% | 5.13% |
| | Avg. | 6.09% | 10.14% | 4.32% | 6.45% | 7.79% | 8.67% | 5.18% | 3.80% | |

Table 10: Model- and tokenizer-wise averages for all error types.

| Error Type | GPT | M1 | M2 | Char | BPE | AIS | NPBPE60 | NPBPE100 | NPBPE1000 | NPBPE7924 | NPBPE30k |
|---|---|---|---|---|---|---|---|---|---|---|---|
| Syntax: N ring openings have not been closed | 48.32% | 47.58% | 44.76% | 42.74% | 46.72% | 48.38% | 44.85% | 45.72% | 48.70% | 51.42% | 46.60% |
| Syntax: Unmatched close parenthesis | 14.82% | 13.20% | 18.18% | 17.11% | 11.75% | 15.57% | 13.90% | 12.59% | 11.26% | 16.87% | 24.18% |
| Kekulization: Aromatic system cannot be kekulized | 13.33% | 14.50% | 12.25% | 14.80% | 13.84% | 15.66% | 13.16% | 15.52% | 14.81% | 10.80% | 8.30% |
| Syntax: N branches have not been closed | 8.62% | 4.30% | 5.84% | 5.33% | 7.62% | 7.41% | 7.78% | 5.56% | 4.13% | 5.57% | 6.63% |
| Valence: Uncommon valence or charge state | 5.13% | 7.64% | 6.89% | 6.09% | 10.14% | 4.32% | 6.45% | 7.79% | 8.67% | 5.18% | 3.80% |
| Syntax: Cannot have a second bond between the same atoms | 4.39% | 4.97% | 4.56% | 4.60% | 3.77% | 3.75% | 4.00% | 5.04% | 5.52% | 5.10% | 5.34% |
| Syntax: Illegal character | 1.34% | 1.92% | 1.73% | 3.15% | 1.65% | 1.76% | 3.60% | 2.25% | 0.60% | 0.18% | 0.12% |
| Syntax: Ring closure symbols must immediately follow an atom | 1.01% | 1.63% | 1.44% | 0.84% | 0.34% | 0.92% | 1.17% | 1.53% | 2.52% | 1.87% | 1.67% |
| Syntax: Missing the close bracket | 0.66% | 0.92% | 0.97% | 1.81% | 2.23% | 0.00% | 1.38% | 0.86% | 0.43% | 0.06% | 0.03% |
| Syntax: The final branch should not be within parentheses | 0.39% | 0.65% | 0.62% | 0.39% | 0.19% | 0.14% | 0.38% | 0.57% | 0.69% | 0.87% | 1.19% |
| Syntax: An atom must precede an open parenthesis | 0.39% | 0.38% | 0.44% | 0.50% | 0.21% | 0.23% | 0.48% | 0.33% | 0.58% | 0.39% | 0.54% |
| Syntax: Cannot have a bond opening and closing on the same atom | 0.37% | 0.59% | 0.54% | 0.67% | 0.30% | 0.61% | 0.81% | 0.68% | 0.46% | 0.26% | 0.20% |
| Syntax: Only a single bond symbol should be used | 0.33% | 0.49% | 0.49% | 0.35% | 0.33% | 0.53% | 0.45% | 0.52% | 0.56% | 0.43% | 0.35% |
| Syntax: Ring closure symbols should not be in parentheses | 0.31% | 0.46% | 0.47% | 0.50% | 0.30% | 0.41% | 0.65% | 0.37% | 0.43% | 0.35% | 0.29% |
| Syntax: An atom must precede a bond symbol | 0.17% | 0.14% | 0.14% | 0.13% | 0.04% | 0.05% | 0.12% | 0.13% | 0.17% | 0.23% | 0.34% |
| Syntax: Empty branches are not allowed | 0.13% | 0.20% | 0.18% | 0.31% | 0.14% | 0.11% | 0.36% | 0.13% | 0.10% | 0.12% | 0.08% |
| Syntax: A bond symbol should not precede an open parenthesis | 0.08% | 0.12% | 0.16% | 0.12% | 0.10% | 0.08% | 0.14% | 0.16% | 0.15% | 0.12% | 0.11% |
| Syntax: An atom must follow a bond symbol | 0.06% | 0.11% | 0.13% | 0.13% | 0.08% | 0.02% | 0.15% | 0.13% | 0.11% | 0.09% | 0.09% |
| Syntax: An element symbol is required | 0.06% | 0.08% | 0.10% | 0.32% | 0.09% | 0.00% | 0.10% | 0.07% | 0.05% | 0.01% | 0.01% |
| Syntax: An atom must precede a bond closure symbol | 0.06% | 0.07% | 0.07% | 0.04% | 0.11% | 0.06% | 0.03% | 0.03% | 0.06% | 0.06% | 0.14% |
| Syntax: An open square brackets is present without the corresponding close square brackets | 0.01% | 0.02% | 0.03% | 0.06% | 0.01% | 0.00% | 0.03% | 0.02% | 0.02% | 0.00% | 0.00% |

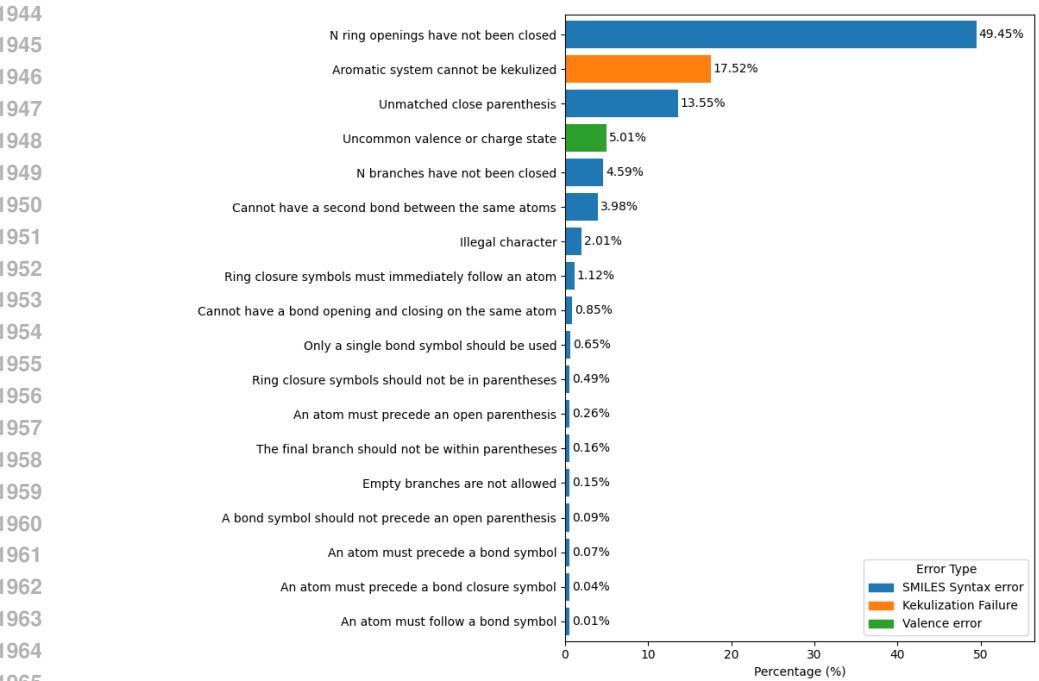

Figure 21: Error type distribution of generated pseudo-NP SMILES (Model: M1-AIS-rds).

### A.10 SCAFFOLDS

Scaffolds are the core frameworks of molecules, and they are essential for classifying compounds and predicting biological activity (Bemis & Murcko, 1996). Thus, their investigation facilitates the identification of privileged substructures and provides insights that are essential for drug development (Hu et al., 2016; 2011; Sato et al., 2021). The scaffold distribution in the 1M NP dataset follows Zipf's law precisely (see Figure 22), meaning the most frequent scaffold appears approximately twice as often as the second most frequent, three times as often as the third, and so forth. A visualization of the most commonly occurring scaffolds in 1M NPs is provided in Figure 24.

As shown in Table 2 in the main text, tokenizers with more fine-grained tokens, like Character-level and NPBPE60, appear to perform better when generating novel scaffolds, contrasting with the earlier observation that the atom-level AIS tokenizer is more effective for generating entire molecules. Scaffolds perhaps benefit from finer-grained tokenization because they involve core substructures that recur with subtle variations, and fine-grained tokens offer such flexibility; however, a better understanding of what underlies this phenomenon might require more chemical and domain-specific expertise.

Regarding scaffold diversity within each model-tokenizer pair's generated molecules, we can observe that they are able to mimic the scaffold diversity present in the training data very well, as they closely resemble the 1M NP's scaffold Zipfian distribution (see Figure 22). The total number of unique and novel scaffolds also suggests that all novel scaffolds lie in the long tail of the Zipfian distributions since these novel scaffolds appear fewer than 10 times each in every model-generated dataset, as shown in Figures 25 and 26.

In addition, Figure 23 shows low pairwise Jaccard similarity values (0.05 to 0.11) between each two sets of scaffolds, suggesting that different models are exploring fairly distinct chemical spaces. Notably, NPBPE30k appears more distinct from other tokenizers while not exhibiting high overlap with the training data, possibly due to its already generating fewer unique scaffolds in the first place.

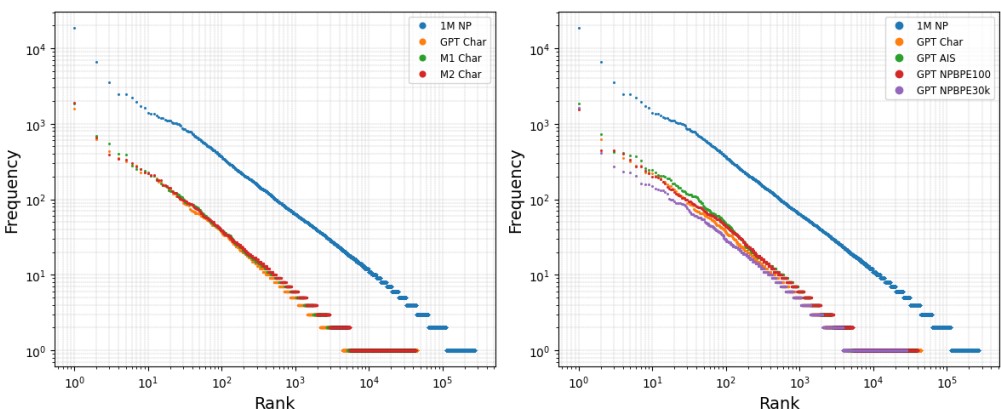

Figure 22: **Zipf's law in scaffold rank-frequency distributions across datasets,** the log-log rank-frequency distributions of all scaffolds of the generated molecules (generated by selected random split pre-trained models) vs. the 1M NP dataset; the x-axis denotes the rank of each unique scaffold based on its occurrence, while the y-axis represents its frequency.

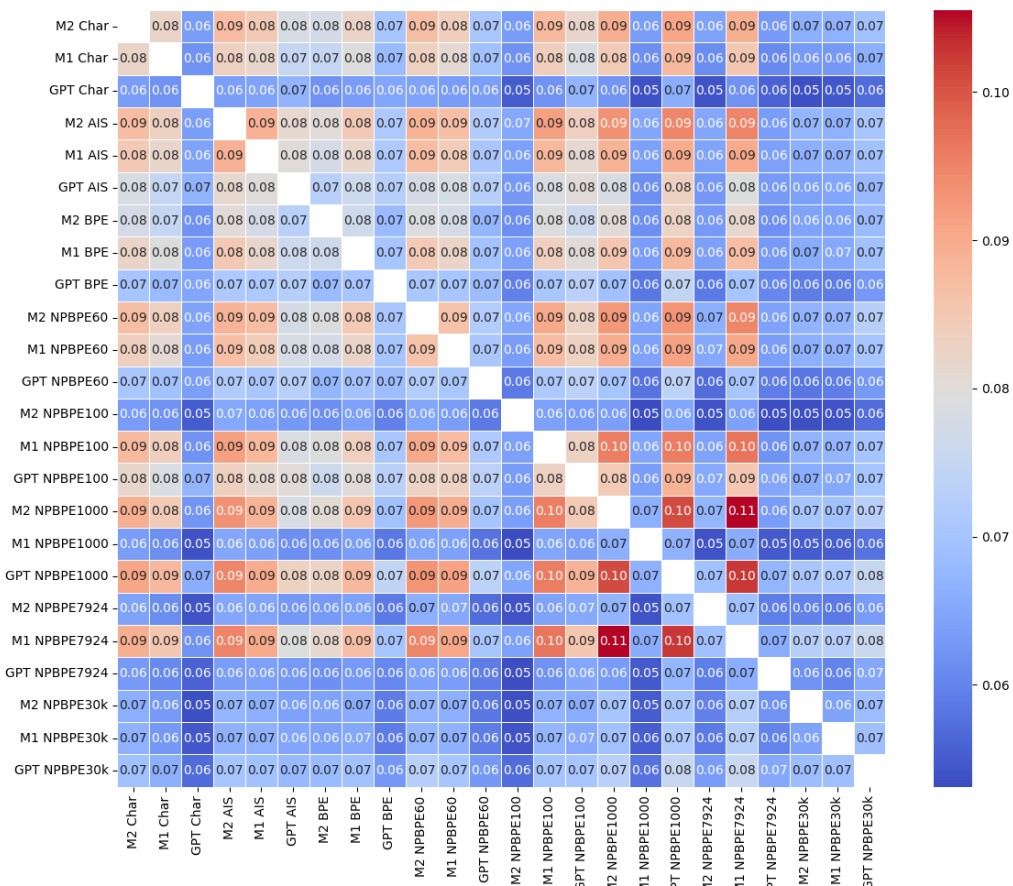

Figure 23: **Jaccard similarity heatmap of scaffold sets.** The heatmap visualizes the pairwise Jaccard similarity between scaffold sets (without duplicates, generated by random split pre-trained models); diagonal self-similarity values are excluded; the color scale represents the similarity values, with warmer colors indicating higher similarity; since the unique scaffold counts for each set range from approximately 30,000 to 45,000, a difference of 0.01 in Jaccard similarity corresponds to an overlap of approximately 550 to 850 scaffolds.

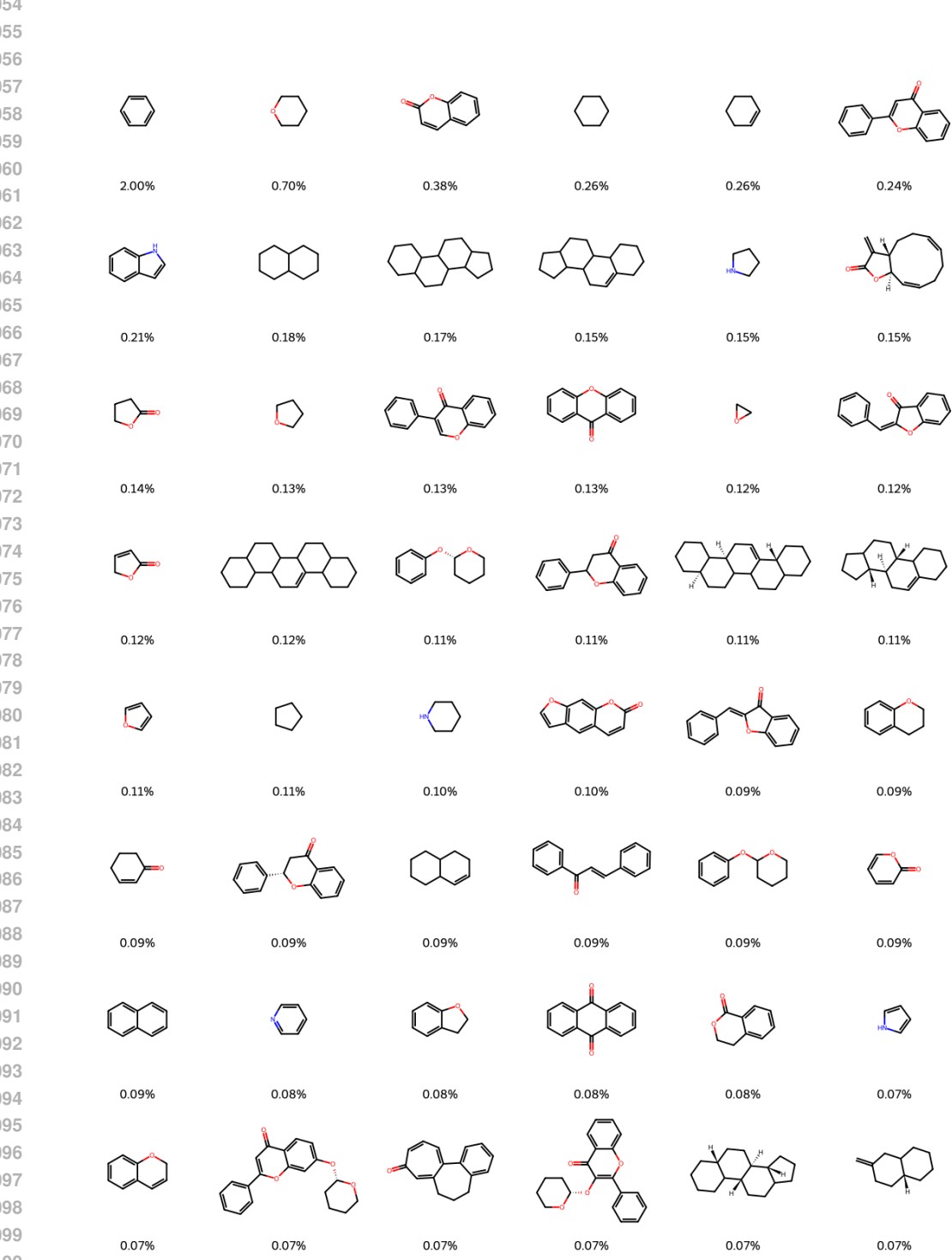

Figure 24: Top 48 most common scaffolds in 1M NPs. Percentages indicate the frequency of each scaffold's occurrence.

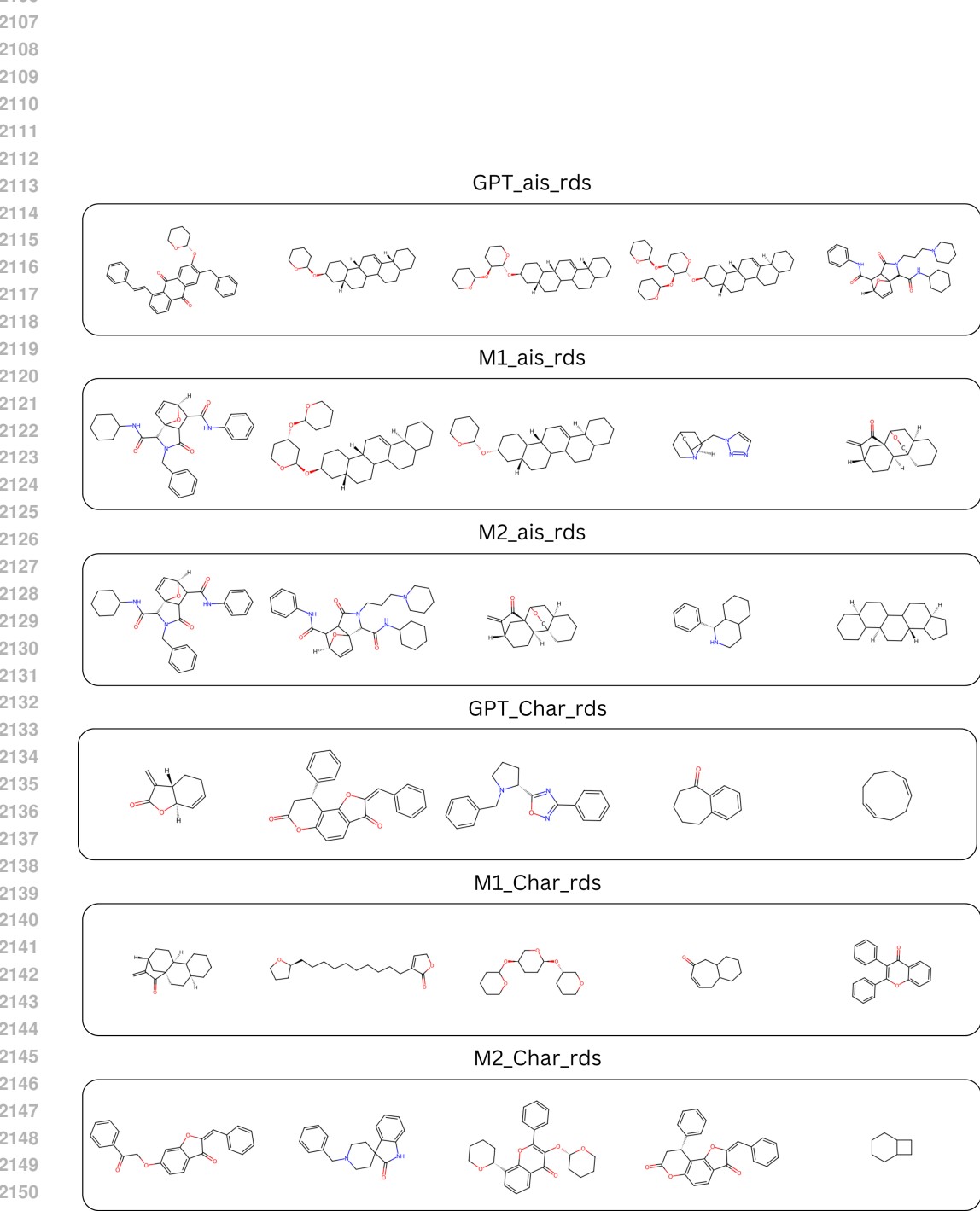

Figure 25: Top 5 most frequently occurring novel scaffolds from molecules generated by selected models - part 1 (with frequencies around 0.01%–0.02%).

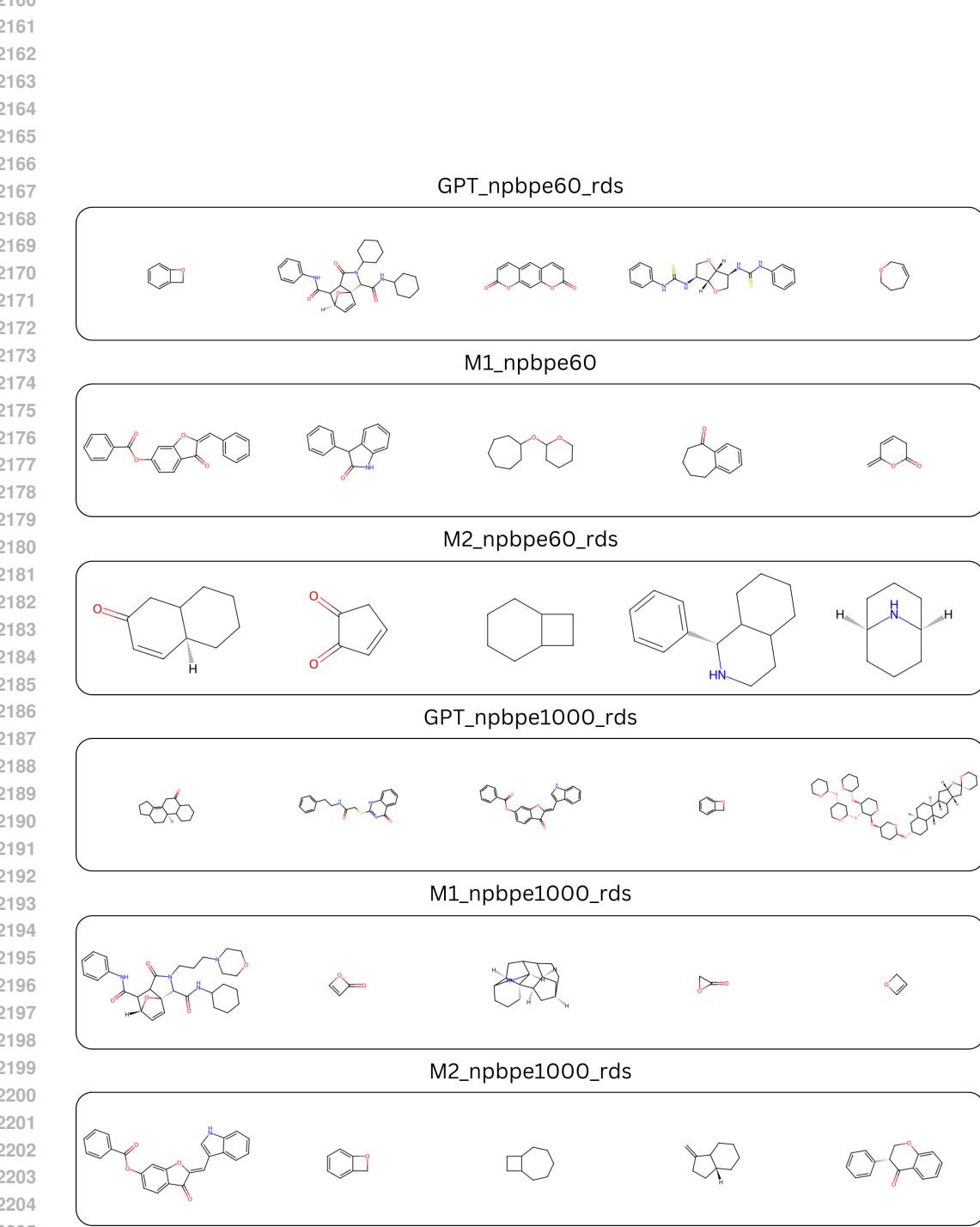

Figure 26: Top 5 most frequently occurring novel scaffolds from molecules generated by selected models - part 2 (with frequencies around 0.01%–0.02%).

## A.11 SYNTHETIC ACCESSIBILITY & NP-LIKENESS

The NP-likeness score and SAScore are widely used cheminformatics metrics developed over a decade ago to evaluate NP similarity and molecule synthetic accessibility. **NP-likeness score**, developed in 2008 by Ertl et al. (2008), is a naive Bayesian-based measure designed to evaluate how structurally similar a given molecule is to known NPs. It has been trained on approximately 115,000 NPs and nearly 300,000 synthetic molecules from commercial libraries, assigning fragment-based scores that reflect structural similarity to known NPs. The score can range from -5 to 5, with higher values indicating greater structural similarity to NPs. **Synthetic accessibility score (SAScore)**, developed by Ertl & Schuffenhauer (2009), assesses the ease of synthesizing drug-like molecules by combining fragment frequency (from PubChem) with penalties for structural complexity, outputting a score from 1 (easy) to 10 (hard). While both metrics are fast and interpretable, they rely on potentially outdated fragment libraries and limited validation, risking poor generalization to novel NPs or synthetically tractable but complex molecules. For completeness, we report both metrics in Table 11, and in Figures 19 and 20; however, we ultimately consider them limited and outdated for the robust evaluation of machine-generated pseudo-NPs that have learned from the more modern NP compound libraries.

Several retrosynthesis tools, such as AiZynthFinder, Chemformer, IBM RXN, Merck Synthia, and ASKCOS, offer automated synthesis route planning and have been successfully applied to a wide range of small synthetic molecules. However, these models are ultimately not designed for NPs, which have more structural complexity. For completeness, we have applied ASKCOS[19] to 25 de novo generated pseudo-NP SMILES strings and reported four resulting retrosynthesis trees (Figure 27) (Tu et al., 2025). In most cases, the model failed to identify valid synthetic routes, highlighting limited applicability to novel NP-like scaffolds. When routes can be found, they are generally plausible; however, many precursor compounds received strongly negative scores, suggesting they are rare, costly, or synthetically impractical.

While synthetic accessibility is an important consideration in drug discovery, it is ultimately not the primary focus of this work. NPs themselves are often synthetically challenging, and this complexity can only increase in machine-generated pseudo-NPs that explore new regions of chemical space. This work centers on evaluating state-space models and tokenizer combinations for NPCLMs. A thorough assessment of synthetic accessibility for pseudo-NPs will be pursued in future work.

Table 11: **Tokenizer- and model-wise averaged NP-likeness and SAScore**: combining both scaffold and random split pre-trained model generation results.

|  | NP-likeness | SAScore |
|---|---|---|
| 1M NP Avg. | 1.23 | 4.47 |
| *Tokenizer-wise Avg.* | | |
| Character-level | 1.09 | 4.20 |
| AIS | 1.11 | 4.20 |
| BPE | 1.10 | 4.19 |
| NPBPE60 | 1.08 | 4.22 |
| NPBPE100 | 1.14 | 4.25 |
| NPBPE1000 | 1.13 | 4.25 |
| NPBPE7924 | 1.08 | 4.20 |
| NPBPE30k | 1.00 | 4.10 |
| *Model-wise Avg.* | | |
| Mamba-2 | 1.11 | 4.18 |
| Mamba | 1.12 | 4.21 |
| GPT | 1.05 | 4.21 |

---

[19]https://askcos.mit.edu/

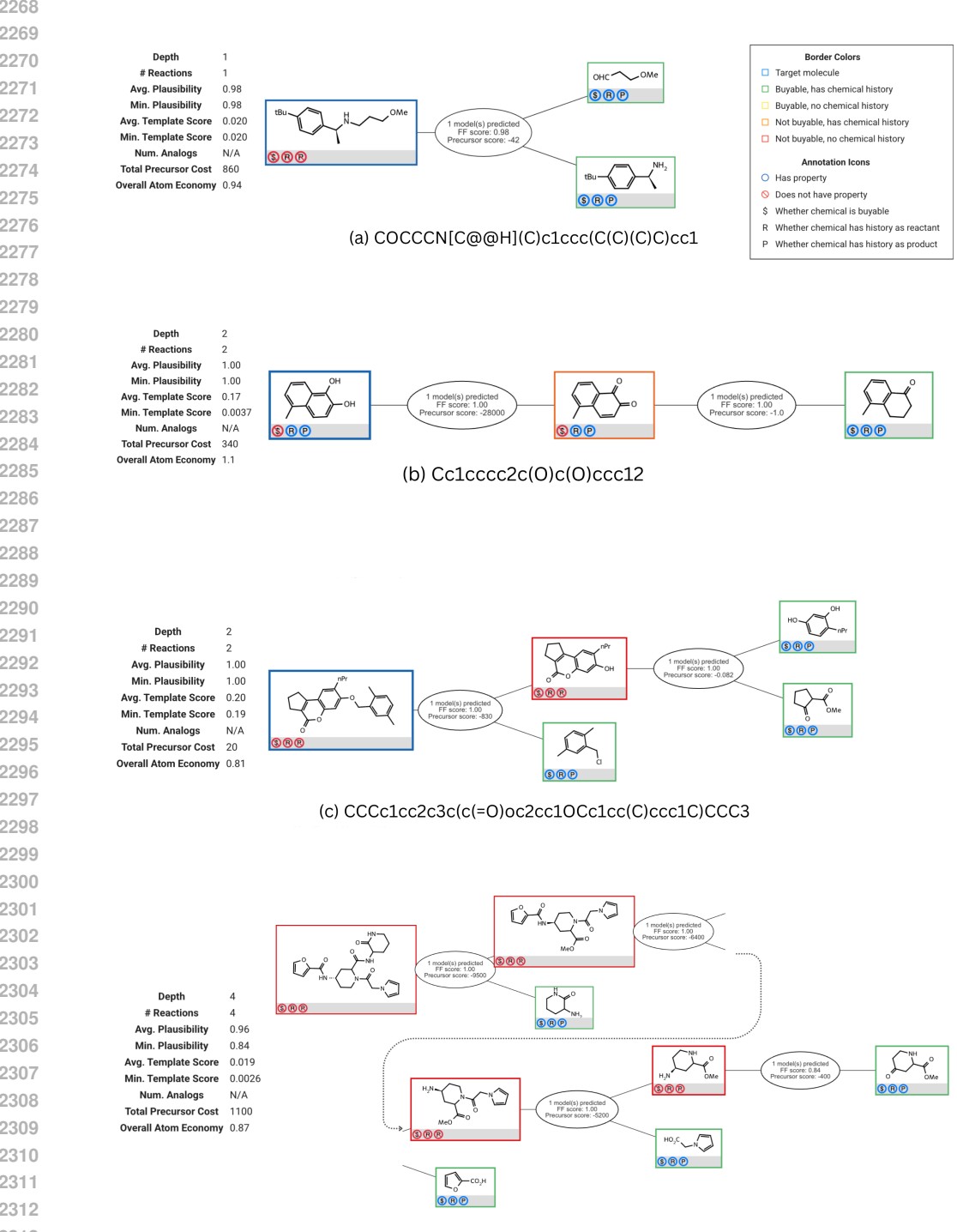

Figure 27: Retrosynthetic pathway analyses of representative pseudo-NP molecules generated by the M1-ais-rds model. Retrosynthetic trees were constructed using ASKCOS (backend: template_relevance; tree builder: MCTS).

## A.12 PRE-TRAINING DETAILS

Before training begins, 5% of the training and validation data is used for a random search that explores half of all possible combinations for the best hyperparameters for each model and tokenizer pair. The hyperparameter search space is shown in Table 12. There are 48 model variations, consisting of 24 model-tokenizer pairs (3 model types times 8 tokenizers) across two data-splitting strategies: scaffold split and random split. Tokenized SMILES sequences are truncated or padded to a fixed length of 512 tokens for batch processing, and cross-entropy loss is used as the objective function. After the hyperparameter search is completed, each model is initialized with its optimal hyperparameters and trained until convergence. The maximum number of training epochs for each model is 150, and training stops if the validation loss does not improve for 5 subsequent epochs. All models in this study have converged before reaching 150 epochs. After training, the final evaluation is performed on the test set that has not been seen by the models during training. A100 GPUs are used for all training processes, and the total training time taken for each model to converge ranges from a few hours to around a week. Figures 28, 29, and 30 show the pre-training details for each model and tokenizer combination in this study.

Sharpness Aware Minimization (SAM) is used as an optimization strategy during training to improve model generalization (Foret et al., 2020). The code implementation in this work is adapted from the work of Tsai et al. (2024) in their open source repository[20]. SAM can improve model generalization by simultaneously minimizing both the value of the loss function and the sharpness of the loss landscape. It is motivated by the idea that flatter minima in the loss landscape lead to better generalization in overparameterized models. Rather than only focusing on a low-loss parameter configuration, SAM seeks parameter regions where the loss remains low across a neighborhood. Implementing SAM requires calculating the loss and performing backpropagation twice per iteration—first to estimate an adversarial perturbation and then to compute the final gradient—which doubles the computational cost compared to standard methods.

Table 12: Pre-training hyperparameter search space.

| Hyperparameters | Mamba and Mamba-2 | GPT |
|---|---|---|
| Number of Mamba Blocks | 4, 8, 12 | n.a. |
| Number of Transformer Blocks | n.a. | 4, 8, 12 |
| Number of Attention Heads | n.a. | 2, 4, 8 |
| Embedding Size | 256, 512 | |
| MLP/FF dimension | Embedding Size * 4 | |
| Learning Rate | 1e-3, 5e-3, 1e-4, 5e-4 | |
| Batch Size | 32 | |
| SAM - L2 Weight Decay | 1e-4 | |
| SAM - Perturbation Radius ($\rho$) | 0.05 | |

---

[20]https://github.com/tiffany9056/UU-Mamba.git

**Model: Mamba 2**

**Scaffold Split**

| Model Details | Character-Level (vocab_size = 48) | Atom-in-SMILES (vocab_size = 1023) | BPE (vocab_size = 7924) | NPBPE60 (vocab_size = 60) | NPBPE100 (vocab_size = 100) | NPBPE1000 (vocab_size = 1000) | NPBPE7924 (vocab_size = 7924) | NPBPE30k (vocab_size = 30k) |
|---|---|---|---|---|---|---|---|---|
| Model Name/ Size | M2-Char-sfs 58.4M | M2-AIS-sfs 59.4M | M2-BPE-sfs 66.5M | M2-npbpe60-sfs 58.5M | M2-npbpe100-sfs 58.5M | M2-npbpe1000-sfs 59.4M | M2-npbpe7924-sfs 66.5M | M2-npbpe30k-sfs 50.2M |
| Best Hyper-Parameters | d_model: 512 n_layer: 12 lr: 0.0001 | d_model: 512 n_layer: 12 lr: 0.0001 | d_model: 512 n_layer: 12 lr: 0.0001 | d_model: 512 n_layer: 12 lr: 0.0005 | d_model: 512 n_layer: 12 lr: 0.0005 | d_model: 512 n_layer: 12 lr: 0.0001 | d_model: 512 n_layer: 12 lr: 0.0001 | d_model: 512 n_layer: 4 lr: 0.001 |
| Test Loss and Perplexity | Loss: 0.5335 Perplexity: 1.7048 | Loss: 0.6210 Perplexity: 1.8608 | Loss: 0.7123 Perplexity: 2.0386 | Loss: 0.7558 Perplexity: 2.1294 | Loss: 1.2084 Perplexity: 3.3482 | Loss: 2.3314 Perplexity: 10.2922 | Loss: 3.6402 Perplexity: 38.1004 | Loss: 5.6934 Perplexity: 296.9032 |
| Training Time | 6d 18hr 13min (2hr50min/epoch) | 2d 6hr 5min (2hr50min/epoch) | 3d 17hr (2hr15min/epoch) | 2d 9hr 4min (1hr50min/epoch) | 1d 9hr 5min (1hr20min/epoch) | 23hr 30min (1hr10min/epoch) | 1d 3hr 5min (55min/epoch) | 18hr 15min (30min/epoch) |
| Inference Time (10k Molecule Generation) | 6hr 4min | 3hr 23min | 5hr 2min | 3hr 50min | 1hr 13min | 34min | 27min | 8min |

**Random Split**

| Model Details | Character-Level (vocab_size = 48) | Atom-in-SMILES (vocab_size = 1023) | BPE (vocab_size = 7924) | NPBPE60 (vocab_size = 60) | NPBPE100 (vocab_size = 100) | NPBPE1000 (vocab_size = 1000) | NPBPE7924 (vocab_size = 7924) | NPBPE30k (vocab_size = 30k) |
|---|---|---|---|---|---|---|---|---|
| Model Name/ Size | M2-Char-rds 58.4M | M2-AIS-rds 59.4M | M2-BPE-rds 27.6M | M2-npbpe60-rds 58.5M | M2-npbpe100-rds 39M | M2-npbpe1000-rds 59.4M | M2-npbpe7924-rds 47M | M2-npbpe30k-rds 69.7M |
| Best Hyper-Parameters | d_model: 512 n_layer: 12 lr: 0.0001 | d_model: 512 n_layer: 12 lr: 0.0001 | d_model: 512 n_layer: 4 lr: 0.0001 | d_model: 512 n_layer: 12 lr: 0.0001 | d_model: 512 n_layer: 8 lr: 0.0001 | d_model: 512 n_layer: 12 lr: 0.0001 | d_model: 512 n_layer: 8 lr: 0.0005 | d_model: 512 n_layer: 8 lr: 0.0005 |
| Test Loss and Perplexity | Loss: 0.2977 Perplexity: 1.3468 | Loss: 0.3479 Perplexity: 1.4161 | Loss: 0.4089 Perplexity: 1.5052 | Loss: 0.3837 Perplexity: 1.4677 | Loss: 0.8388 Perplexity: 2.3136 | Loss: 1.3155 Perplexity: 3.7267 | Loss: 2.7081 Perplexity: 15.0005 | Loss: 3.6911 Perplexity: 40.0905 |
| Training Time | 8d 8hr 11min (2hr50min/epoch) | 4d 18hr 12min (2hr15min/epoch) | 4d 21hr (1hr/epoch) | 4d 17hr 20min (2hr11min/epoch) | 1d 23hr 17min (50min/epoch) | 3d 4hr 35min (1hr/epoch) | 21hr 15min (40min/epoch) | 1d 10hr 1min (39min/epoch) |
| Inference Time (10k Molecule Generation) | 7hr 39min | 4hr 9min | 1hr 59min | 4hr 16min | 1hr 15min | 45min | 23min | 15min |

Figure 28: Mamba-2 training details.

| Model | Data Split | Model Details | Character-Level (vocab_size = 48) | Atom-in-SMILES (vocab_size = 1023) | BPE (vocab_size = 7924) | NPBPE60 (vocab_size = 60) | NPBPE100 (vocab_size = 100) | NPBPE1000 (vocab_size = 1000) | NPBPE7924 (vocab_size = 7924) | NPBPE30k (vocab_size = 30k) |
|---|---|---|---|---|---|---|---|---|---|---|
| Mamba 1 | Scaffold Split | Model Name/Size | M1-Char-sfs 58.1M | M1-AIS-sfs 39.8M | M1-BPE-sfs 66.2M | M1-npbpe60-sfs 58.2M | M1-npbpe100-sfs 58.2M | M1-npbpe1000-sfs 39.8M | M1-npbpe7924-sfs 8.96M | M1-npbpe30k-sfs 69.5M |
| | | Best Hyper-Parameters | d_model: 512 n_layer: 12 lr: 0.0001 | d_model: 512 n_layer: 8 lr: 0.0001 | d_model: 512 n_layer: 12 lr: 0.0001 | d_model: 512 n_layer: 12 lr: 0.0001 | d_model: 512 n_layer: 12 lr: 0.0001 | d_model: 512 n_layer: 8 lr: 0.0005 | d_model: 256 n_layer: 4 lr: 0.0005 | d_model: 512 n_layer: 8 lr: 0.0005 |
| | | Test Loss and Perplexity | Loss: 0.5317 Perplexity: 1.7018 | Loss: 0.6288 Perplexity: 1.8754 | Loss: 0.7108 Perplexity: 2.0356 | Loss: 0.6920 Perplexity: 1.9978 | Loss: 1.1526 Perplexity: 3.1663 | Loss: 2.4916 Perplexity: 12.0805 | Loss: 3.8768 Perplexity: 48.2701 | Loss: 5.3646 Perplexity: 213.7134 |
| | | Total Training Time | 4d 17hr 9min (2hr45min/epoch) | 1d 23hr 25min (1hr40/epoch) | 4d 3hr 5min (2hr35/epoch) | 2d 4hr 26min (2hr50/epoch) | 15hr 20min (1hr15/epoch) | 11hr 30min (30min/epoch) | 8hr 27min (12min/epoch) | 14hr 7min (25min/epoch) |
| | | Inference Time (10k Molecule Generation) | 6hr 23min | 2hr 13min | 4hr 52min | 3hr 52min | 1hr 15min | 18min | 7min | 11min |
| | Random Split | Model Name/Size | M1-Char-rds 38.8M | M1-AIS-rds 59.1M | M1-BPE-rds 66.2M | M1-npbpe60-rds 9.83M | M1-npbpe100-rds 58.2M | M1-npbpe1000-rds 39.8M | M1-npbpe7924-rds 46.9M | M1-npbpe30k-rds 69.5M |
| | | Best Hyper-Parameters | d_model: 512 n_layer: 8 lr: 0.0001 | d_model: 512 n_layer: 12 lr: 0.0001 | d_model: 512 n_layer: 12 lr: 0.0001 | d_model: 256 n_layer: 8 lr: 0.0001 | d_model: 512 n_layer: 12 lr: 0.0001 | d_model: 512 n_layer: 8 lr: 0.0005 | d_model: 521 n_layer: 8 lr: 0.0001 | d_model: 512 n_layer: 8 lr: 0.0005 |
| | | Test Loss and Perplexity | Loss: 0.3041 Perplexity: 1.3554 | Loss: 0.3552 Perplexity: 1.4265 | Loss: 0.4013 Perplexity: 1.4937 | Loss: 0.3959 Perplexity: 1.4856 | Loss: 0.6500 Perplexity: 1.9156 | Loss: 1.7016 Perplexity: 5.4828 | Loss: 2.0349 Perplexity: 7.6517 | Loss: 3.6825 Perplexity: 39.7459 |
| | | Total Training Time | 7d 12hr 10min (2hr10min/epoch) | 7d 9hr 24min (2hr10min) | 7d 13hr 44min (2hr50min/epoch) | 1d 21hr 9min (34min/epoch) | 6d 21hr 46min (1hr10/epoch) | 1d 1hr 48min (37min/epoch) | 2d 4hr 14min (30min/epoch) | 1d 1hr 5min (30min/epoch) |
| | | Inference Time (10k Molecule Generation) | 5hr | 3h 29min | 5hr 17min | 1hr 20min | 1hr 27min | 23min | 14min | 10min |

Figure 29: Mamba training details.

| Model | Data Split | Model Details | Character-Level (vocab_size = 48) | Atom-in-SMILES (vocab_size = 1023) | BPE (vocab_size = 7924) | NPBPE60 (vocab_size = 60) | NPBPE100 (vocab_size = 100) | NPBPE1000 (vocab_size = 1000) | NPBPE7924 (vocab_size = 7924) | NPBPE30k (vocab_size = 30k) |
|---|---|---|---|---|---|---|---|---|---|---|
| GPT | Scaffold Split | Model Name/Size | GPT-Char-sfs 25.5M | GPT-AiS-sfs 13.9M | GPT-BPE-sfs 33.6M | GPT-npbpe60-sfs 25.5M | GPT-npbpe100-sfs 13M | GPT-npbpe1000-sfs 13.9M | GPT-npbpe7924-sfs 21M | GPT-npbpe30k-sfs 56.2M |
| | | Best Hyper-Parameters | n_embd: 512 n_layer: 8.0 n_head: 4.0 lr: 0.0001 | n_embd: 512 n_layer: 4 n_head: 8 lr: 0.0001 | n_embd: 512 n_layer: 8 n_head: 8 lr: 0.0001 | n_embd: 512 n_layer: 8 n_head: 8 lr: 0.0001 | n_embd: 512 n_layer: 4 n_head: 8 lr: 0.0001 | n_embd: 512 n_layer: 4 n_head: 8 lr: 0.0001 | n_embd: 512 n_layer: 4 n_head: 8 lr: 0.0005 | n_embd: 512 n_layer: 8 n_head: 8 lr: 0.001 |
| | | Test Loss and Perplexity | Loss: 0.5971 Perplexity: 1.8169 | Loss: 0.6460 Perplexity: 1.9080 | Loss: 0.7471 Perplexity: 2.1108 | Loss: 0.7269 Perplexity: 2.0686 | Loss: 1.1808 Perplexity: 3.2571 | Loss: 2.3754 Perplexity: 10.7556 | Loss: 4.0420 Perplexity: 56.9387 | Loss: 5.8333 Perplexity: 341.4829 |
| | | Total Training Time | 1d 22hr 33min (1hr15min/epoch) | 1d 21hr 51min (50min/epoch) | 2d 6hr 41min (1hr30/epoch) | 2d 4hr 25min (1hr/epoch) | 7hr 47min (15min/epoch) | 7hr 22min (10min/epoch) | 6hr 11min (9min/epoch) | 5hr 27min (16min/epoch) |
| | | Inference Time (10k Molecule Generation) | 4hr 12min | 1hr 11min | 3hr 55min | 2hr 32min | 20min | 7min | 10min | 12min |
| | Random Split | Model Name/Size | GPT-Char-rds 25.5M | GPT-AiS-rds 13.9M | GPT-BPE-rds 33.6M | GPT-npbpe60-rds 12.9M | GPT-npbpe100-rds 13M | GPT-npbpe1000-rds 13.9M | GPT-npbpe7924-rds 21M | GPT-npbpe30k-rds 43.6M |
| | | Best Hyper-Parameters | n_embd: 512 n_layer: 8 n_head: 8 lr: 0.0001 | n_embd: 512 n_layer: 4 n_head: 4 lr: 0.0001 | n_embd: 512 n_layer: 8 n_head: 8 lr: 0.0001 | n_embd: 512 n_layer: 4 n_head: 8 lr: 0.0001 | n_embd: 512 n_layer: 4 n_head: 2 lr: 0.0001 | n_embd: 512 n_layer: 4 n_head: 8 lr: 0.0001 | n_embd: 512 n_layer: 4 n_head: 8 lr: 0.0005 | n_embd: 512 n_layer: 4 n_head: 8 lr: 0.0005 |
| | | Test Loss and Perplexity | Loss: 0.3852 Perplexity: 1.4699 | Loss: 0.4166 Perplexity: 1.5168 | Loss: 0.4770 Perplexity: 1.6112 | Loss: 0.4695 Perplexity: 1.5992 | Loss: 0.7624 Perplexity: 2.1435 | Loss: 1.4130 Perplexity: 4.1083 | Loss: 2.9623 Perplexity: 19.3424 | Loss: 3.8917 Perplexity: 48.9935 |
| | | Total Training Time | 3d 13hr 34min (1hr30min/epoch) | 2d 3hr (45min/epoch) | 2d 16hr 26min (1hr15min/epoch) | 1d 12hr (30min/epoch) | 1d 2hr 12min (20min/epoch) | 20hr 3min (11min/epoch) | 6hr 1min (10min/epoch) | 11hr (11min/epoch) |
| | | Inference Time (10k Molecule Generation) | 4hr 36min | 1hr 8min | 3hr 22min | 1hr 15min | 27min | 8min | 16min | 9min |

Figure 30: GPT training details.

