# OpenReview forum: "Chemical Language Models for Natural Products: A State-Space Model Approach"
_ICLR.cc/2026/Conference — Submitted to ICLR 2026_

### Official Review · Reviewer_8tF2 · 2025-10-20

**Soundness:** 2
**Presentation:** 3
**Contribution:** 2
**Rating:** 4
**Confidence:** 4

**Summary:**

This paper introduces **Natural Product Chemical Language Models (NPCLMs)**, addressing the underrepresentation of natural products (NPs) in molecular language modeling. Using a curated dataset of ~1M NP SMILES, the authors pre-train state-space models (Mamba, Mamba-2) and compare them with transformer baselines (GPT), systematically evaluating eight tokenization strategies. Experiments span **molecule generation** (validity, uniqueness, novelty, scaffold diversity) and **property prediction** (anti-cancer activity, taste, and peptide permeability). Results show that **Mamba** yields more valid and unique molecules with fewer long-range syntax errors, while **GPT** generates slightly more novel scaffolds. For property prediction, Mamba and Mamba-2 modestly outperform GPT under random splits but converge under scaffold splits. Crucially, **tokenization choice** strongly impacts outcomes, with smaller, chemically informed vocabularies (AIS, NP-specific BPE) outperforming large vocabularies. The study highlights that domain-specific pretraining on NPs can match or exceed general chemical LMs trained on 100× more data, emphasizing the importance of data quality and relevance over sheer scale.

**Strengths:**

1. **Essential field and motivation**: The paper addresses an important and underexplored problem: developing chemical language models specifically for natural products (NPs). Since NPs are chemically more complex and biologically significant than typical synthetic molecules, focusing on them fills a meaningful gap in current research and aligns well with drug discovery applications. By targeting this space, the paper contributes to an essential field with high scientific and practical relevance.
2. **Systematic and comprehensive experiments**: The study is notable for its breadth and rigor in experimental design. It benchmarks three different model families (Mamba, Mamba-2, GPT) across eight tokenization strategies, under both random and scaffold splits, using a curated dataset of ~1M NPs. The authors evaluate on molecule generation and property prediction tasks with multiple metrics, conduct error analysis, and even compare with general CLMs like ChemBERTa-2 and MoLFormer. This systematic approach makes the results highly credible and useful as a benchmark reference.
3. **Clear writing and presentation**: The paper is well-written and accessible, with a clear motivation, logical structure, and smooth narrative flow from background to experiments to conclusions. The figures and tables effectively summarize key findings, while detailed reproducibility notes in the appendices strengthen transparency. Overall, the writing quality and presentation make the technical contributions easy to follow for both machine learning researchers and domain experts in chemistry.

**Weaknesses:**

The main limitation of the paper is that the contribution largely reduces to an extensive set of empirical comparisons, but without introducing new methodological innovations or deriving deeper insights that could guide future work.

1. **Model comparison (Mamba vs GPT)**: The comparison between Mamba and GPT models offers limited value. The results mainly show that Mamba tends to generate molecules with higher validity and uniqueness, while GPT produces more novel ones. However, this trade-off between validity and novelty is easily tunable through standard decoding hyperparameters (temperature, top-k, top-p) in both model families, and outcomes are also highly sensitive to training settings such as the number of steps. As a result, the comparison does not provide clear guidance on which architecture is better suited for molecular tasks.
2. **Tokenizer comparison**: The tokenizer study similarly contributes little novelty. Prior work [1] has already shown the superiority of atom-level tokenization over generic BPE approaches. Moreover, the models here are trained only on canonical SMILES without augmentation by randomized SMILES, which is known to benefit large-vocabulary BPE tokenizers. This design choice biases the results and weakens the conclusions.
3. **Connection to natural products**: Although the paper positions itself around natural products, the methods and analyses are only weakly tied to NP-specific characteristics. The dataset merely provides a source of training molecules, and the main conclusions (validity vs novelty trade-off, tokenizer effects) appear general to SMILES modeling rather than unique to natural products.
4. **Choice of architectures for property prediction**: While autoregressive models are natural for molecule generation, they are not state-of-the-art for molecular property prediction, where graph neural networks and 3D-aware models currently dominate. The paper’s detailed comparison of autoregressive architectures in this context therefore seems of limited relevance.

[1] Comparing SMILES and SELFIES tokenization for enhanced chemical language modeling.

**Questions:**

1. **Motivation for NP-specific training**: Could the authors elaborate on the motivation for training exclusively on natural products? Intuitively, pretraining on a broader dataset that includes but is not limited to NPs might improve generalization, while still capturing NP-specific features. Why was NP-only pretraining chosen instead of a mixed strategy?
2. **Validity of generated molecules**: In Table 1, the reported validity ratios are relatively low (even with atom-level tokenization). Prior works on SMILES generation typically report validity above 90% [1]. Could the authors clarify why validity is lower here?
3. **Effect of model scaling**: What is the impact of scaling model size in this setting? Given that GPT-type transformers often benefit strongly from scaling, do the authors expect GPT to outperform Mamba under larger-scale setups?

[1] Molecular de-novo design through deep reinforcement learning.

---

> ### Author Response · Authors · 2025-11-22
>
> The reviewer is concerned about the limited conceptual contribution. Our work, however, provides a systematic benchmark for (i) selective SSMs vs. transformers, (ii) tokenization strategies, and (iii) domain-specific vs. broad pre-training in the underexplored NP regime, across both generation and property prediction. This kind of controlled, architecture–tokenizer–data comparison is currently missing in CLM work on NPs and more broadly, in dense symbolic sequences, despite strong interest in applying Mamba-like models to chemistry and biology.
>
> **1. Model comparison (Mamba vs GPT)**
>
> The reviewer notes that validity/novelty trade-offs can be tuned via decoding hyperparameters. We agree that temperature/top-k/top-p can shift this trade-off for any model family. In our study, we intentionally fix decoding (temperature = 1, unconstrained sampling) to probe intrinsic tendencies under matched conditions. These systematic differences, together with prior evidence that S4/Mamba better stabilizes long-range dependencies while attention promotes flexible recombination, suggest that the contrast comes from the models’ internal mechanisms, not from superficial decoding settings.
>
> Furthermore, Our model comparison is not meant to crown a single “better” architecture, but to give practitioners guidance on which model family tends to favor validity/consistency vs exploration/novelty in dense chemical sequences. **As we note in the paper, specific model–tokenizer combinations are better suited for different goals. In practice, there is no single architecture that is universally ‘best’ for molecular tasks—any such claim would oversimplify a complex landscape. The appropriate choice always depends on the intended objective.**
>
> **2. Tokenizer comparison and relation to prior work**
>
> Tokenization for SMILES has been studied before, but not extensively. Prior work has shown the benefits of atom-aware or motif-based tokenization (e.g., SMILES Pair Encoding, AIS) over naïve character/BPE but only for general small-molecule CLMs. **Our contribution is to systematically compare tokenizers within the Mamba/Mamba-2/GPT families, do so specifically for the NP regime rather than generic drug-like space, and evaluate on both generation and downstream property prediction.**
>
> The reviewer mentions that “atom-level better than generic BPE” is known. **We indeed reproduce this, but with two new insights**:
> 1. We show that **when the vocabulary size is tuned properly, BPE can work as well as an atom level tokenizers**.
> 2. We show that **tokenization preferences differ for molecules vs scaffolds**: AIS/NPBPE60–100 are best for full-molecule validity/novelty, but finer-grained tokenization (character-level, NPBPE60) yields more novel scaffolds, which is not captured in earlier work.
>
> Regarding **randomized SMILES augmentation**, we intentionally avoid it. **Although it can act as a useful regularizer, it also distorts the natural data distribution and complicates controlled comparisons, since tokenizers may benefit unevenly—large-vocabulary BPE tends to gain far more than atom-level schemes.** It can also inflate performance by adding many near-duplicate sequences, increase training cost through dataset expansion, and hinder reproducibility unless all seeds and generation settings are fixed. For work focused on isolating architectural or tokenizer effects, these drawbacks outweigh the benefits.
>
> **3. Connection to NPs**
>
> Correct us if we’re wrong, but **we believe that the reviewer’s critique here rests on a category error and a shifting standard: it assumes that a study is “not really about natural products” unless every method and finding is unique to NPs and cannot appear in any other chemical domain. But domain specificity is defined by the data, motivations, and analyses—which in this case are entirely NP-focused**—not by whether some modeling behaviors (e.g., tokenizer effects or validity–novelty trade-offs) also arise in other SMILES settings. The work uses NP-only data, NP-relevant downstream tasks, and analyses grounded in NP structural properties; just as a study on bird flight is still about birds even if some principles also apply to bats or airplanes.
>
> This expectation—that NP-focused work must yield findings unique to NPs in order to count as genuinely NP-specific—misframes the issue, because the study’s NP focus comes from the domain, data, and structural motivations, while the fact that some behaviors generalize simply reflects properties of sequence models, not a lack of NP relevance.

---

> > ### Author Response · Authors · 2025-11-22
> >
> > **4. Choice of architectures for property prediction (AR vs GNN/3D)**
> >
> > We agree that GNNs and 3D-aware models set the state of the art for certain molecular property prediction benchmarks. **Our focus here is deliberately narrower: we compare sequence-based CLMs, which remain widely used in industrial pipelines** (as practitioners have shared with us privately). Including graphs and 3D baselines would significantly broaden the scope and make it difficult to attribute effects to architecture vs representation. We therefore treat GNN/3D models as complementary rather than competing with the main question of “within sequence models, what are the trade-offs between selective SSMs, transformers, and tokenization for dense NP sequences?”. We will make this framing clearer and highlight the explicit direction to incorporate non-LM baselines for follow-up work.

---

> ### Author Response · Authors · 2025-11-22
>
> **Responses to specific questions**
>
> **Q1** Motivation for NP-specific training vs broader pretraining.
>
> **Our central objective is to test, in a controlled way, whether pretraining exclusively on NPs is sufficient for strong performance on NP-focused tasks, or whether exposure to broader molecular domains is actually necessary.** To isolate this effect, we train models from scratch solely on NPs, and compare them directly against other large, broadly trained chemical language models—ChemBERTa-2 (77M SMILES from diverse sources) and MoLFormer (≈110M molecules from PubChem+ZINC)—both with and without additional NP-specific fine-tuning. Our results show that NP-only Mamba models match or slightly surpass these much larger, broadly pretrained CLMs on anti-cancer and peptide-permeability tasks, despite using 77–110× less pretraining data. **This provides clear evidence that NP-restricted pretraining is not only viable but can rival broad-domain pretraining for NP-enriched applications. We agree that broad pretraining followed by NP-focused adaptation are intuitively appealing. However, their effectiveness could not be assumed without empirical evidence.** Our results show that NP-only and mixed pretraining occupy distinct but complementary regions of the design space, and our study provides a controlled, well-defined reference point within that space.
>
> **Q2** Validity of generated molecules (<90% vs typical reports).
>
> The reviewer notes that our average validity (≈75–81%) is below some reported values (>90%). **We believe the reviewer 8tF2’s comparison to Olivecrona et al.’s work from 2017 here is inappropriate because their model operates on a broad, drug-like chemical space rather than NPs solely. Their Prior RNN is pretrained on 1.5 M RDKit-canonicalized ChEMBL molecules—structures that are far smaller, simpler, and more homogeneous than NPs—and already achieves ~94% validity before any additional training. Their reinforcement-learning setup also explicitly penalizes invalid SMILES, further inflating validity.** In contrast, our models are trained solely on structurally diverse NPs without a SMILES-optimized Prior or validity-based rewards. NP SMILES are inherently harder to model due to more complex stereochemistry, larger scaffolds, and less regular ring systems. The lower validity we observe reflects the intrinsic difficulty of NP generation and our choice to keep the decoding setup deliberately simple to enable fair comparisons across models. **If one’s primary goal is to maximize validity, this is straightforward to achieve, for example, by lowering the sampling temperature or applying a validity filter during generation, doing so can easily push validity close to 100%. This point is unrelated to any model or methodological deficiency, and especially should not be compared directly to models trained on a fundamentally different, less complex chemical space.**
>
> If a comparison is needed, an NP-focused example is NPGPT [1], which reports validity values of 99% and 90%. Their high validity is expected given their setup: they use a much smaller NP set (around 400k molecules), expand it almost ten-fold via SMILES enumeration, and fine-tune models already pretrained on PubChem-10M, which have effectively learned SMILES syntax in advance. This combination of low data diversity, heavy augmentation, and strong pretrained priors naturally inflates syntactic validity. By contrast, our models are trained on a substantially larger and far more diverse NP corpus (around 1M NPs) without relying on any augmentation or SMILES-optimized pretraining, which makes the task more challenging and yields lower, but more representative, validity values.
>
> [1] Sakano et al. NPGPT: natural product‑like compound generation with GPT‑based chemical language models
>
> **Q3** Effect of model scaling and expectations for larger GPT vs Mamba.
>
> We agree that our study does not systematically vary model size, and we explicitly list this as a limitation. **GPT models are known to benefit strongly from scaling, but so do Mamba models.** Both Mamba and Mamba-2 have been reported to maintain competitive accuracy across model sizes compared to Transformer models, as shown in [1] & [2].  We considered predicting which architecture would dominate at much larger parameter counts beyond the scope of this study. Our results should be interpreted as comparisons within the model sizes; we do not claim that the relative ordering would remain unchanged under substantially larger-scale setups.
>
> [1] Gu & Dao (2023) — Mamba: Linear-Time Sequence Modeling with Selective State Spaces
> [2] Dao & Gu (2024) — Transformers Are SSMs: Generalized Models and Efficient Algorithms Through Structured State Space Duality (Mamba-2)
>
> We thank the reviewer again for the time and thoughtful assessments.

---

### Official Review · Reviewer_fJft · 2025-10-30

**Soundness:** 2
**Presentation:** 3
**Contribution:** 1
**Rating:** 2
**Confidence:** 5

**Summary:**

The work compares three different model architectures on the generation and property prediction of natural products. Each architecture is combined with eight different tokenizers and two different train-test splits (random and scaffold). The results are presented systematically and align with the previous findings in the literature.

I believe that the paper should be rejected.
1. Neither of the applied architectures nor the tokenizers is novel. SSMs are already used on natural product generation (Ozcelik 2025) and the architectures tested here display no significant improvement over prior work.
2. The findings largely revalidate the literature, offering limited new insights.
3. More controls and evaluation are needed to support the significance of the findings.
4. The application domain, natural products, is a narrow field for the ICLR community.

**Strengths:**

1. A large and systematic comparison was conducted.
2. The presentation and results discussion are detailed and easy to follow.
3. The covered related work is comprehensive and well-aligned with the discussion..
4. Mamba architectures are applied to natural product modeling for the first time.

**Weaknesses:**

**Significance**
1. The focus of the work is natural products, which is a narrow subfield of drug discovery. The significance of the findings to the broader ICLR community is limited. The work can better fit technical and specialized venues of drug discovery.
2. Mamba and some tokenizers are applied to natural products for the first time here. However, these approaches perform similarly to the existing work and only confirm the findings (as cited multiple times by the authors), yielding no new insights for the community.

**Quality**

3. S4s are missing in the comparisons. A work that focuses on SSMs and natural products cannot ignore S4s, which are the first SSMs used in this domain. A non-language model baseline (such as www.nature.com/articles/s42004-023-01054-6) should also be included to better contextualize the performance of the architectures.
4. The authors use validity, novelty, or uniqueness to compare the models for generation. Yet, these metrics only measure the syntactic performance. More metrics, such as descriptor similarity, diversity, NP-likeness should be added to capture 'semantic' performance.
5. Scaffold-split does not guarantee distance between train and test sets -- non-identical scaffolds can still be very similar (arxiv.org/abs/2406.00873). A distance-based split should be applied to better capture generalization performance.
6. Hyperparameters are tuned on 5 percent of the data and used for training on the entire dataset. This is a jump from small-scale to large-scale training, and can lead to suboptimal performance for each architecture, jeopardizing the model-wise comparison.

**Questions:**

1. What is the motivation behind measuring validity, uniqueness, and novelty with different train/test splits? There are unconditional generation metrics that are independent of the test set. The results also show no real influence.
2. What are the novel findings of the work? The tokenizers comparison, error analysis, impact of the pre-training set, and model-wise comparison all largely confirm prior findings in the literature.

---

> ### Author Response · Authors · 2025-11-22
>
> We thank the reviewers for the constructive feedback. Below, we address the main concerns and clarify several points where our original presentation may have been insufficient.
>
> **1. Significance of Natural Products for ICLR**
>
> Reviewer fJft is concerned that NPs are viewed as a narrow subfield with limited relevance for the broader ICLR community. We appreciate this perspective and agree that our framing could better highlight the broader relevance. While NP-focused research is indeed specialized, it reflects the same trajectory many now-central ICLR areas once followed, such as early NLP or protein modeling. Recent ICLR papers on MS/MS-guided molecule generation, antibiotic-focused multimodal models, 3D molecular diffusion, conformer-ensemble benchmarks, molecular docking, bias-corrected molecular representation learning, explainable molecular generation, and 3D molecule–text alignment show that highly focused chemical and bio-molecular tasks are considered meaningful testbeds for modern generative and representation models [1][2][3][4][5][6][7][8]. NP chemistry contains unusually information-dense sequences and highly irregular structural patterns that are not necessarily long—conditions that stress-test sequence models in ways few existing benchmarks do. We will clarify these connections in the revision to better situate our contribution.
>
> ICLR 2025 Poster:
>
> [1] MADGEN: Mass-Spec attends to De Novo Molecular generation (https://openreview.net/forum?id=78tc3EiUrN)
> [2] CL-MFAP: A Contrastive Learning-Based Multimodal Foundation Model for Molecular Property Prediction and Antibiotic Screening (https://openreview.net/forum?id=fv9XU7CyN2)
>
> ICLR 2024 Poster:
> [3] Navigating the Design Space of Equivariant Diffusion-Based Generative Models for De Novo 3D Molecule Generation (https://openreview.net/forum?id=kzGuiRXZrQ)
> [4] Removing Biases from Molecular Representations via Information Maximization (https://openreview.net/forum?id=7TOs9gjAg1)
> [5] Equivariant Scalar Fields for Molecular Docking with Fast Fourier Transforms (https://openreview.net/forum?id=BIveOmD1Nh)
> [6] Beam Enumeration: Probabilistic Explainability For Sample Efficient Self-conditioned Molecular Design (https://openreview.net/forum?id=7UhxsmbdaQ)
> [7] Towards 3D Molecule-Text Interpretation in Language Models (https://openreview.net/forum?id=xI4yNlkaqh)
> [8] Learning Over Molecular Conformer Ensembles: Datasets and Benchmarks (https://openreview.net/forum?id=NSDszJ2uIV)

---

> > ### Author Response · Authors · 2025-11-22
> >
> > **2. “No New Insights” and Similarity to Existing Work**
> >
> > Reviewer fJft is concerned that Mamba and the tokenizers perform “similarly to existing work” and only confirm prior findings. However, we respectfully disagree that our results merely restate prior work. Our experiments yield only NP- and Mamba- specific outcomes that have not been reported elsewhere. **Findings of related work are cited only to indicate directional parallels, not equivalence**, and this, in our view, is a positive sign of robustness.
> > Specialized domains often behave differently from broad ones, showing that a general trend still holds—or breaks—in a narrower, more chemically complex regime is itself new evidence. Even when results are not surprising, they are not conclusions one can assume without empirical verification. Establishing these patterns for NPs—even when they only modestly adjust prior expectations—prevents redundant future work and provides a clear methodological baseline for this emerging area.
> >
> > If the reviewer can kindly point to specific passages and cited works in which the paper claims equivalence with prior work's findings, we would be grateful. Here, we’ve listed several citation in the paper that may have caused such misunderstanding:
> >
> > **(a)** “This aligns with the findings from Aksamit et al. (2024), which show that overly large vocabularies lead to sparse token usage and hinder efficient chemical language modeling.”
> > Aksamit et al. show that very large fragment vocabularies create many rare tokens, leading to sparse statistics and weaker ADMET prediction—a primarily statistical explanation. Our findings point in the same broad direction but for a different reason: large BPE-style vocabularies disrupt chemically meaningful motifs, distort SMILES syntax, and increase structural errors during NP generation. These effects produce drops in validity/uniqueness/novelty that cannot be attributed to sparsity alone. **Crucially, Aksamit et al. do not use BPE; their vocabulary refers to fragment dictionaries.** We will clarify this difference in mechanism.
> >
> > **(b)** Line 300: “This aligns with the expected trade-off in Mamba-2 between inference expressivity and hardware efficiency”
> > This statement reflects a hypothesis from the Mamba authors in their blog about Mamba models (“Empirically, we haven’t found evidence that the restricted expressivity of Mamba-2 might hurt, but the jury’s still out!” - https://tridao.me/blog/2024/mamba2-part1-model/). To our knowledge, ours is the first empirical evaluation of this expressivity–efficiency trade-off of Mamba 2. We will note this context explicitly.
> >
> > **(c)** Line 461: Effect of broad vs. domain-specific pre-training
> >
> > Prior work highlights that broad pre-training can introduce irrelevant features in specialized scientific domains. Our contribution is complementary and more specific: we show that broad molecular pre-training does not improve NP generation and can introduce noise. Before this study, the intuitive assumption—echoed by Reviewer 8tF2—was that “using more data that also include NPs must be better,” and that NPs need not be singled out. **Our results show the opposite: NP-only pre-training is beneficial, and adding broader molecule coverage does not help NP modeling. This also motivates the need for future NP-focused datasets, which are currently lacking.** This finding has not been made elsewhere. We will refine the text to make this contribution clearer.
> >
> > **(d)** On the claim that our findings “confirm prior work” on tokenizers comparison, error analysis, impact of the pre-training set, and model-wise comparison
> >
> > We respectfully note that several components of our work have limited or no precedent:
> > No prior systematic tokenizer comparisons for Mamba/Mamba-2, nor for NPs.
> > Only one prior error-analysis study on generated pseudo-NPs exists [1], but what they compare differs from ours. They evaluate S4, GPT, and LSTM models, use a much smaller NP set (32,360 vs. our >1 million), and NPs are not the main focus of their work. Our analysis covers GPT and Mamba-1/2 and provides a more detailed breakdown of NP-specific error types.
> > No prior comparison of Mamba, Mamba-2, and GPT within chemical language modeling.
> >
> > [1] Chemical language modeling with structured state space sequence models

---

> ### Author Response · Authors · 2025-11-22
>
> **3. Missing S4 Baseline and Non-LM Baseline**
>
> We appreciate this concern. However, **our study focuses on selective SSMs, a category S4 does not belong to**, and prior NP work using S4 was trained on only ~32k NP molecules [1]. Our setup—pre-training on 1M NPs and optimizing for A100 efficiency—led us to concentrate on Mamba and Mamba-2, which are both selective SSMs and are expected to be substantially faster and more memory-efficient than S4 at this scale. Given these practical and conceptual differences, we believe excluding S4 is a reasonable choice for this work, and we will clarify this rationale more explicitly.
>
> Regarding a non-LM baseline, we agree that this is a useful direction. **Our focus here is, however, on sequence-based language models, which remain widely used in industrial pipelines** (as practitioners have shared with us privately). In practice, **sequence models are not replaced by graph models but used alongside them**, so understanding their behavior remains valuable. While graph baselines are important, adding them here would greatly broaden the scope beyond our central question on chemical “LLM”.
>
> [1] Chemical language modeling with structured state space sequence models
>
> **4. Metrics Beyond Validity/Novelty/Uniqueness**
>
> We agree and apologize for not highlighting more clearly what is already included.
> **NP-likeness and generated molecules’ scaffold diversity are provided in the Appendix (A.8–A.11).** We will make these more visible in the main paper. Descriptor-based similarity was not included because comparing two collections of roughly 60–80k molecules each would require computing all pairwise distances, an O(n^2) operation that becomes prohibitively expensive at this scale. Performing the analysis on only a small subset would substantially reduce the computational burden, but it would also risk giving an incomplete or misleading picture of how the two molecular sets differ. We will explain this practical trade-off in more detail.
>
> **5. Scaffold Split vs. Distance-Based Split**
>
> Scaffold splitting does not guarantee large train–test distances, indeed, and thanks for pointing us to distance-based alternatives. **We use the Bemis–Murcko scaffold split because it remains the most common protocol in molecular ML and offers chemical interpretability, reproducibility, and comparability across studies.** While Tossou et al. (2024) [1] emphasize that scaffold splits alone do not capture full real-world OOD behaviour, they still induce a clearer distribution shift than random splits, and Niu et al. (2024) [2] likewise treat them as a standard, more challenging benchmark. In our case, **we already observe a consistent performance drop from random to scaffold splits, indicating that the split meaningfully stresses generalization**. Implementing and validating an additional distance-based split is not central to our goal—the split serves only as a controlled variable, and evaluating splitting strategies is out of scope for this study. We will, however, make this point clearer by stating the limitations of scaffold splitting and acknowledging distance-based alternatives.
>
> [1] Tossou et al., 2024: Real-World Molecular Out-Of-Distribution: Specification and Investigation
> [2] Niu et al., 2024: PharmaBench: Enhancing ADMET benchmarks with large language models

---

> > ### Author Response · Authors · 2025-11-22
> >
> > **6. Hyperparameter Tuning on 5% of Data**
> >
> > We do not believe this setup jeopardizes the model-wise comparison for three reasons.
> >
> > **(1)** The comparison remains valid because all models use the same search protocol, and each model type has eight tokenizer variants. This provides ample independent runs, making it unlikely that the findings are driven by chance or are sensitive to the specific hyperparameter search setup.
> >
> > Even if the absolute optimum for each architecture might shift slightly when tuning on the full dataset, the relative ranking is stable so long as (i) the search space is identical, (ii) the objective is identical, and (iii) the proxy dataset is drawn from the same distribution. **Using a smaller, representative subset prevents leakage and aligns with multi-fidelity HPO, which uses reduced-budget evaluations (e.g. smaller data subset) to efficiently filter configurations and avoid overfitting to potential noise from high-fidelity estimates [1].** Since every Mamba/Mamba-2/GPT configuration is tuned under the same conditions, the comparison remains fair and internally consistent. Tuning directly on 1M NPs makes hyperparameter selection highly data (test split)-dependent: architectures can overfit to the idiosyncrasies of the full pre-training set as well. This risks making the comparison less fair, because one architecture may benefit more from dataset-specific tuning than another.
> >
> > **(2)** The empirical behavior of the models suggests stability across regimes.
> >
> > Across random vs. scaffold splits, and across eight tokenizers, the observed model-wise trends (e.g., Mamba’s validity/uniqueness advantage vs. GPT’s novelty advantage) remain consistent. This stability is strong evidence that the trends are due to architectural differences rather than artifacts of using only 5% subset for hyperparameter tuning.
> >
> > **(3)** Using a smaller subset for HPO  is a practical necessity and aligned with community practice.
> >
> > Running HPO for 48 model–tokenizer pairs would require ~480 full-scale searches—computationally infeasible given that Mamba and GPT also use partially different search spaces. Prior CLM work routinely adopts similar compromises: ChemBERTa tuned hyperparameters on its smallest subset (5M) before training on the full 77M corpus; MoLFormer and several other CLM papers do not specify using the full dataset for hyperparameter search, and it is not standard to assume that full-dataset HPO was performed unless explicitly stated. Our choice, therefore, reflects both feasibility and norms in existing CLM research.
> >
> > We will clarify these points in the revision, while also acknowledging that full-dataset HPO may modestly improve absolute performance but is unlikely to alter the relative conclusions.
> >
> > [1] Won et al., 2025 A review on multi-fidelity hyperparameter optimization in machine learning
> >
> > **7. Motivation for Using Multiple Train/Test Splits in Generation Metrics**
> >
> > Our reason for reporting the same unconditional metrics under both random and scaffold splits is simply that:
> > Evaluating the same generation metrics under both regimes allows us to check whether a harder, scaffold-disjoint split alters model behavior.
> > The fact that the metrics are relatively stable across splits is itself a (non-obvious) empirical finding: despite a harder pre-training split, Mamba/GPT/tokenizer trade-offs largely persist.
> > We will clarify this motivation and make the interpretation explicit in the text.
> >
> > Closing
> > We respect the reviewer’s perspective and understand their interest in a broader set of comparisons. Nevertheless, one challenge in addressing this review is that **fully covering all possible baselines suggested by the reviewer**—multiple selective (Mamba 1&2) plus non-selective (S4) SSMs, transformers, and non-LM graph models—**on top of our goal to include tokenizers comparison, would effectively require a study far broader than the scope of a single paper.** We understand the motivation for comprehensive comparisons, and we agree that these directions are valuable. **At the same time, our goal here is more focused: to provide the first systematic evaluation of selective SSMs and tokenizers for NPs sequence models at scale.** We will clarify this scope explicitly to align expectations. We thank the reviewers again for the thoughtful assessments.

---

> > > ### Comment · Reviewer_fJft · 2025-11-24
> > > **Respond to Author Comments**
> > >
> > > 1. **Significance of Natural Products for ICLR**: I definitely agree that molecules are meaningful testbeds for deep learning paradigms. However, conceiving a method and testing on a high-importance molecular type or on various molecular modalities are different types of works from analyzing existing ideas on a single molecular entity with limited interest. I believe that while the works cited fall into the first category, this work is more of the second type. I am still of the opinion that this work is better suited for a more specialized venue, such as JCIM, as [reviewer kuwX](https://openreview.net/forum?id=TcmsiSkN5T&noteId=CzA7OdsAmj) also suggested.
> > >
> > > 2. **Novelty:** I admit that my initial statement, "no new insights," was an unfair overstatement. Thanks for correcting me. However, the findings are still largely unsurprising and refining the literature in the narrow field of natural products. In other words, all the conclusions cited (a,b,c,d) are either confirmed once more or refined for natural products. Yes, this is still of scientific value, but I believe the contribution is limited for ICLR.
> > >
> > > 3. **S4 and Grap Baseline:** I disagree with this argument. S4s can definitely be trained on 1M NPs feasibly, and they are valid baselines. If the work focuses on Selective SSMs, comparing them with S4s can reveal the importance of selectivity in the architecture. Same for graph-based models. Without comparing against these recent baselines, it is hard to conclude whether Mambas are "the way to go" for future NP research.
> > >
> > > 4. **Metrics Beyond Validity/Novelty/Uniqueness:** I thank the authors for pointing out the appendix. Going forward, I suggest including these semantic metrics in the main text, as they are more meaningful than the syntactic ones, which can easily be "hacked" even by trivial baselines such as ["AddCarbon"](https://www.sciencedirect.com/science/article/pii/S1740674920300159). I disagree with the infeasibility aspect. Distributions of chemical descriptors can be computed set-to-set (no pairwise computation needed) and similarity of designs to the training sets can be computed via subsets. Having a subset-based result is better than not having these metrics at all.
> > >
> > > 5. **Scaffold Split vs Distance-based Split:** I agree that comparing splitting strategies is out-of-scope. I recommend, however, being careful with the generalization claims in the current form, and use scaffold-split only as a more difficult split than random-split, which still needs to be supported by distance measurements.
> > >
> > > 6. **HP Tuning:** Yes, the HP tuning is fair at the current form. I believe authors have done the best possible in their computational environment. Although the model rankings might change with more extensive tuning, I believe the current results are still meaningful to the community.
> > >
> > > 7. **Motivation for measuring syntactic metrics with multiple folds:** I believe there is a misunderstanding here. The metrics I cited (validity, uniqueness, novelty) are independent of the test set. They are measured only over the designs -- with no consideration of a test set. That is why it is hard to motivate measuring them with different train/test splits and expecting changes across different splitting strategies. Put another way, given a fixed pretraining set and one easy and one hard test set, these metrics would be identical across test sets.
> > >
> > > I thank the authors for their response and clarifications. However, I still believe my initial assessment stands. I believe the work is valuable and can be published in a more specialized venue, with a few more experiments.

---

### Official Review · Reviewer_kuwX · 2025-11-01

**Soundness:** 2
**Presentation:** 2
**Contribution:** 2
**Rating:** 4
**Confidence:** 4

**Summary:**

The paper introduces a specialized chemical language model for natural products. The model is based on state-space architecture. The model is benchmarked against transformer baselines.

**Strengths:**

The contribution addresses the gap in modeling natural products in chemistry.

The presentation is clear, the methodology is sound.

**Weaknesses:**

Aside from targeting narrow yet important chemical domain, such as the natural products, the paper does not report any particularly interesting results.

It's a solid research that would be best suited for a cheminformatic journal (Journal of Chemical Information and Modeling or something similar).

**Questions:**

Considering the reasons why state-space models are introduced, please provide some comparison of MAMBA and GPT inference speed and scaling with the sequence size.

---

> ### Author Response · Authors · 2025-11-22
>
> We thank the reviewer for the thoughtful review and appreciate the balanced assessment.
>
> **Inference speed and scaling**
>
> We do report inference-time comparisons between Mamba/Mamba-2 and GPT in the main text (Lines 304–318) and in the Appendix (Figures 28–30).
>
> **Scaling with sequence length**
>
> We did not emphasize this experiment because NP SMILES are typically medium to short in length, and our aim is to model this distribution faithfully. Evaluating very long-sequence scaling would require stepping outside the NP regime or artificially separating NPs by length, which we believe would not align with the paper's scientific focus. We appreciate the suggestion, however, and agree that long-sequence scaling is an interesting direction for follow-up work.
>
> We thank the reviewer again for their constructive feedback. We understand the concern about venue fit and appreciate the suggestion of specialized cheminformatics outlets.

---

### Meta-Review · Area_Chair_3RHT · 2025-12-31

**Summary:**

This paper introduces Natural Product Chemical Language Models (NPCLMs), addressing a critical gap in computational chemistry by developing specialized language models for natural products (NPs) using state-space models (Mamba, Mamba-2) and comparing them with transformer baselines. While the paper is a systematic and well-presented empirical study that addresses a clear gap in modeling the Natural Product domain, it fails to meet the bar for technical novelty and broader significance required for ICLR. The reviewers consistently pointed out that the findings largely revalidate existing literature or offer only incremental refinements specific to a narrow subfield. Furthermore, the lack of essential baselines, such as non-selective SSMs (S4) or SOTA graph-based models, makes it difficult to determine if the proposed Mamba-based approach offers a genuine advantage over current established methods.

**Reviewer Concerns:**

Addressed concerns:
 - Performance metrics (Reviewer fJft): The authors clarified that semantic metrics like NP-likeness and scaffold diversity were indeed calculated and present in the Appendix.
 - Hyperparameter tuning (Reviewer fJft): The reviewer accepted the authors' explanation that tuning on 5% of the data was a fair and practical necessity given the computational environment.
 - LLM fairness (Reviewer 8tF2): The authors provided a robust defense of their validity scores, explaining that NPs are structurally more complex than the simpler drug-like molecules used in prior studies that reported >90% validity.

Outstanding concerns:
 - Venue fit (Reviewers kuwX, fJft): Multiple reviewers maintain that this work, while solid, is better suited for specialized cheminformatics venues (like JCIM) rather than a general machine learning conference.
 - Technical novelty (Reviewers fJft, 8tF2, kuwX): The fundamental concern remains that applying existing architectures (Mamba/GPT) and known tokenization strategies to a new (though important) dataset does not constitute a significant methodological contribution.
 - Missing baselines (Reviewer fJft): The refusal to include S4 or graph-based models remains a point of contention. Reviewer fJft explicitly disagreed with the authors, noting that S4 is a feasible and necessary baseline to justify the importance of "selectivity" in the architecture.
 - Intrinsic model differences (Reviewer 8tF2): The concern that the validity-novelty trade-off is merely a byproduct of decoding hyperparameters rather than architectural superiority remains unmitigated.

**Reviewer Scores:**

- Reviewer kuwX (original score: 4): Would likely maintain at 4.
 - Reviewer fJft (original score: 2): Would likely maintain at 2.
 - Reviewer 8tF2 (original score: 4): Would likely maintain at 4.

---

### Decision · Program_Chairs · 2026-01-26

Reject